# Soft magnetic hysteresis in a dysprosium amide–alkene complex up to 100 kelvin

Jack Emerson-King[1], Gemma K. Gransbury[1], Benjamin E. Atkinson[1,2], William J. A. Blackmore[1], George F. S. Whitehead[1], Nicholas F. Chilton[1,2 ✉] & David P. Mills[1 ✉]

Lanthanides have shown magnetic memory at both the atomic[1,2] and molecular[3,4] level. The magnetic remanence temperatures of lanthanide single-molecule magnets can surpass $d$-transition metal examples[5,6], and since 2017, energy barriers to magnetic reversal ($U_{eff}$) from 1,237(28) cm$^{-1}$ to 1,631(25) cm$^{-1}$ and open magnetic hysteresis loops between 40 K and 80 K have typically been achieved with axial dysprosium(III) bis(cyclopentadienyl) complexes[7–17]. It has been predicted that linear dysprosium(III) compounds could deliver greater $m_J$ (the projection of the total angular momentum, $J$, on a quantization axis labelled $z$) state splitting and therefore higher $U_{eff}$ and hysteresis temperatures[18–21], but as lanthanide bonding is predominantly ionic[22,23], so far dysprosium bis(amide) complexes have shown highly bent geometries that promote fast magnetic reversal[24,25]. Here we report a dysprosium bis(amide)–alkene complex, [Dy{N(Si$^i$Pr$_3$)[Si($^i$Pr)$_2$C(CH$_3$)=CHCH$_3$]}{N(Si$^i$Pr$_3$)(Si$^i$Pr$_2$Et)}][Al{OC(CF$_3$)$_3$}$_4$] (**1-Dy**), that shows $U_{eff}$ = 1,843(11) cm$^{-1}$ and slow closing of soft magnetic hysteresis loops up to 100 K. Calculations show that the $U_{eff}$ value for **1-Dy** arises from the charge-dense amide ligands, with a pendant alkene taking a structural role to enforce a large N–Dy–N angle while imposing only a weak equatorial interaction. This leads to molecular spin dynamics up to 100 times slower than the current best single-molecule magnets above 90 K.

The dysprosium bis(amide)–alkene complex [Dy{N(Si$^i$Pr$_3$)[Si($^i$Pr)$_2$C(CH$_3$)=CHCH$_3$]}{N(Si$^i$Pr$_3$)(Si$^i$Pr$_2$Et)}][Al{OC(CF$_3$)$_3$}$_4$] (**1-Dy**) and its diamagnetic yttrium analogue **1-Y** were synthesized in 8–13% yields by protonation of the respective lanthanide bis(amide)–allyl complexes [Ln{N(Si$^i$Pr$_3$)[Si($^i$Pr)$_2$C(CH$_3$)CHCH$_2$]}{N(Si$^i$Pr$_3$)(Si$^i$Pr$_2$Et)}] (**2-Ln**; Ln = Dy, Y) with [HNEt$_3$][Al{OC(CF$_3$)$_3$}$_4$] (ref. 12) in benzene at 40 °C for 18 h, followed by recrystallization from fluorobenzene solutions layered with hexane (Fig. 1a). A doped sample **5%Dy@1-Y** was prepared by co-crystallization of a mixture of **1-Dy** and **1-Y**. Inspired by literature protocols[26], **2-Ln** were synthesized in 15–17% yields via the reactions of parent LnI$_3$ with [K{N(Si$^i$Pr$_3$)$_2$}] in benzene at 100 °C (Fig. 1a) as the sole benzene-soluble lanthanide-containing reaction products. Under the forcing reaction conditions employed, an in situ dehydrogenative carbon–carbon (C–C) bond rearrangement of the ligand scaffold had occurred; the mechanism of this transformation will be elucidated in a separate study.

The nuclear magnetic resonance (NMR) spectra of **1-Dy** and **2-Dy** were not fully interpreted owing to paramagnetism (Supplementary Figs. 1–3), but the interatomic connectivity of **1-Y** (Extended Data Fig. 1) and **2-Y** (Supplementary Figs. 4–6) are unambiguous from their NMR spectra. Powder X-ray diffraction (XRD) patterns and infrared spectra for Dy/Y pairs of complexes are consistent with each other (Extended Data Fig. 2, Supplementary Figs. 7–13 and Supplementary Table 1) and single-crystal data. Le Bail profile analysis of the powder XRD data indicates high phase purity in all cases[27]; these data were cross-referenced by elemental analyses.

Single-crystal XRD was performed on all previously unknown complexes to determine their solid-state structures (Fig. 1b, Supplementary Figs. 14–17 and Supplementary Table 2). All datasets required extensive disorder modelling; thus, we do not make firm conclusions on the significance of individual metrical parameters as a range of values is present. Electron density plots show that the models used are appropriate (Supplementary Figs. 18–27). We note that **2-Ln** show similar metrical parameters to other rare earth allyl complexes[28], but otherwise restrict our discussion to the cation of **1-Dy**, which was modelled as two competitively refined components, **1-Dy-A** and **1-Dy-B**, in a 0.649(5):0.351(5) ratio. This cation has Dy–N distances of 2.205(9) Å (**1-Dy-A**) and 2.166(12) Å (**1-Dy-B**) to the tethered amide, 2.217(8) Å (**1-Dy-A**) and 2.236(11) Å (**1-Dy-B**) for the terminal amide, N–Dy–N angles of 150.1(5)° (**1-Dy-A**) and 165.3(8)° (**1-Dy-B**), and twist angles between the two planes described by the Si–N–Si atoms measuring 58.2(3)° (**1-Dy-A**) and 62.0(4)° (**1-Dy-B**). The dysprosium bis(amide) complex [Dy{N(Si$^i$Pr$_3$)$_2$}$_2$][Al{OC(CF$_3$)$_3$}$_4$] has similar Dy–N distances (2.206(7) Å mean), with a more bent N–Dy–N angle (128.7(2)°), and a more pronounced twist angle (71.49(12)°)[25]. The weak η$^2$-alkene binding in **1-Dy** is evidenced by one proximal Dy⋯H$_{alkene}$ distance (2.519 Å for **1-Dy-A**; 2.851 Å for **1-Dy-B**), two short Dy⋯C$_{alkene}$ distances (2.806(16) Å and 2.750(16) Å for **1-Dy-A**; 2.798(17) Å and 3.00(2) Å for **1-Dy-B**), and C=C bond lengths (1.300(16) Å for **1-Dy-A**; 1.337(17) Å for **1-Dy-B**) that are consistent with an unbound C=C double bond (1.34 Å)[29]. Structurally authenticated lanthanide alkene and alkyne complexes are rare[30–34]

[1]Department of Chemistry, The University of Manchester, Manchester, UK. [2]Research School of Chemistry, The Australian National University, Canberra, Australian Capital Territory, Australia. ✉e-mail: nicholas.chilton@anu.edu.au; david.mills@manchester.ac.uk

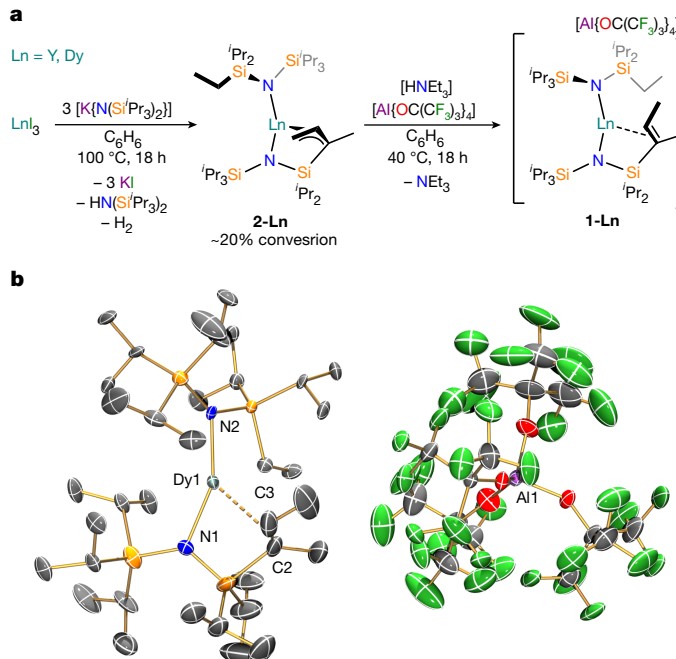

**Fig. 1 | Synthesis and structure of 1-Dy. a**, Synthesis of the lanthanide amide–alkene complexes **1-Ln** (Ln = Dy, Y) in two steps from parent LnI₃, via respective lanthanide amide–allyl complexes **2-Ln**. **b**, Molecular structure of the major component of **1-Dy** at 99.96(18) K with selective atom labelling (Al, purple; C, grey; Dy, cyan; F, green; N, blue; O, red; Si, orange). Displacement ellipsoids set at 30% probability level and hydrogen atoms and disorder components are omitted for clarity. Selected distances (Å) and angles (°): major component (0.649(5)), **1-Dy-A**: Dy1–N1, 2.205(9); Dy1–N2, 2.217(8); Dy1···C2, 2.806(16); Dy1···C3, 2.750(16); C2–C3, 1.300(16); N1–Dy1–N2, 150.1(5). Minor component (0.351(5)), **1-Dy-B**: Dy1–N1, 2.166(12); Dy1–N2, 2.236(11); Dy1···C2, 2.798(17); Dy1···C3, 3.00(2); C2–C3, 1.337(17); N1–Dy1–N2, 165.3(8).

as these show weak electrostatic interactions[28]. Density functional theory (DFT) and quantum theory of atoms in molecules analysis of **1-Y** at an optimized geometry (Supplementary Table 3) confirm that the η²-alkene interaction is weak compared with Y–N interactions (electron density, $\rho$, at the Y–C(alkene) bond critical point is 0.034 a.u. (where a.u. is atomic units), versus $\rho \approx 0.085$ a.u. at the Y–N critical points), although slightly larger than a previously reported Yb–η²-alkyne complex with $\rho \approx 0.016$ a.u. at the DFT-optimized geometry[34]. The coordination spheres in **1-Dy** are completed by two additional short Dy···C contacts (Dy(1)···C(21), 2.943(10) and Dy(1)···C(35), 2.709(14) Å for **1-Dy-A**; Dy(1)···C(21), 2.90(3) and Dy(1)···C(35), 2.81(3) Å for **1-Dy-B**) and two Dy···Si distances <3.3 Å (Dy(1)···Si(1), 3.168(4) Å and Dy(1)···Si(4), 3.144(8) Å for **1-Dy-A**; Dy(1)···Si(1), 3.055(7) Å and Dy(1)···Si(4), 3.193(15) Å for **1-Dy-B**); this leads to an additional three Dy···H distances less than 2.6 Å in each component. Electrostatic interactions between the electron density of the Si–C/C–H bonds of silyl groups and coordinatively unsaturated lanthanide ions are commonplace in *f*-block silylamide chemistry[21,35], for example, [Dy{N(Si^iPr₃)₂}₂][Al{OC(CF₃)₃}₄] has six Dy···H distances less than 2.6 Å (ref. 24). We posit that the increased magnetic anisotropy of **1-Dy** is mainly due to the pendant alkene pinning the coordinated ligand into place, although both intra- and inter-ligand dispersion force interactions[36] and crystal packing forces[37] contribute to its less bent N–Dy–N angle.

Complete active space self-consistent field spin–orbit (CASSCF-SO) calculations were performed using OpenMolcas[38] on both components of the single-crystal XRD structure of **1-Dy** and its DFT-optimized geometry (Fig. 2a and Extended Data Table 1), and **2-Dy** (Supplementary Table 4 and Supplementary Fig. 28); we focus on the results for the

major disorder component (N–Dy–N angle of 150.1(5)°) of **1-Dy** here. The strong crystal field imposed by the two bound amides is evidenced by a large splitting of the electronic states, an essentially pure maximal $m_J = \pm 15/2$ (the projection of the total angular momentum, $J$, on a quantization axis labelled $z$) Kramers doublet ground state with Ising-like magnetic anisotropy (where effective $g$-values in $x$-, $y$- and $z$- Cartesian directions for the ground Kramers doublet are $g_x = g_y = 0$, $g_z = 19.89$), and relatively high purities of the excited states, despite the bent N–Dy–N angle and the transverse field imposed by the tethered alkene (Extended Data Table 1). Magnetic reversal by one-phonon interactions is expected to proceed over a barrier formed by the fourth excited $m_J$ state (where $g_x$ or $g_y > 1$, 1,809 cm⁻¹ above the ground state, 56% $m_J = \pm 7/2$, 40° between excited $g_z$ and ground $g_z$). The crystal field splitting generated by the silylamides in **1-Dy** is greater than previously observed for related axial dysprosium(III) complexes containing aromatic ligands[7–17], for example, [Dy(Cp^{ttt})₂][B(C₆F₅)₄] (Cp^{ttt} = {C₅H₂^tBu₃-1,2,4}; mean Dy···Cp_{centroid}, 2.316(2) Å; Cp_{centroid}···Dy···Cp_{centroid}, 152.56(7)°; energy barriers to magnetic reversal $U_{eff} = 1,237(28)$ cm⁻¹)[7] and [Dy(C₅^iPr₅)(C₅Me₅)][B(C₆F₅)₄] (Dy···Cp_{centroid}, 2.296(1) Å and 2.284(1) Å; Cp_{centroid}···Dy···Cp_{centroid}, 162.51(2)°; $U_{eff} = 1,550(7)$ cm⁻¹)[11]. Although the N–Dy–N angles of **1-Dy-A** and **1-Dy-B** are similar to the corresponding Cp_{centroid}···Dy···Cp_{centroid} angles of these literature complexes, their Dy–N distances are far shorter than the respective Dy···Cp_{centroid} distances and the anionic ligand charges in **1-Dy** are formally located on N atoms rather than π-delocalized; this greater charge density should enhance the crystal field splitting[39]. The $U_{eff}$ of theoretical two-coordinate dysprosium bis-(amide), -(alkyl) and -(methanediide) compounds have been shown to vary substantially with Dy–L distances and L–Dy–L angles, with predicted values exceeding 3,200 cm⁻¹ for linear systems with Dy–L bonds of 2.0 Å (refs. 23,40); a theoretical bent dysprosium bis(aryloxide) cation [Dy(OC₆H₃^tBu₂-2,6)₂]⁺ with Dy–O distances of 2.189(2) Å and a bent O–Dy–O angle of 155.49(5)° has been predicted to show $U_{eff}$ up to 2,286 cm⁻¹ (ref. 41).

The magnetic properties of **1-Dy**, **5%Dy@1-Y**, **2-Dy** and a palladium reference sample were studied on a superconducting quantum interference device (SQUID; Fig. 2b,c, Extended Data Figs. 3–6, Supplementary Figs. 29–41 and Supplementary Tables 5–7). The product of molar magnetic susceptibility ($\chi$) and temperature ($T$), $\chi T$, for **1-Dy** under a 0.1-T direct current (d.c.) field is nearly linear with temperature (Extended Data Fig. 3a,b), consistent with a large magnetic anisotropy and in agreement with the CASSCF-SO-calculated trace. The zero-field-cooled and field-cooled $\chi T$ values diverge below 47 K owing to magnetic blocking[42], which causes $\chi$ and $\chi T$ to drop below the expected equilibrium values. Calibration of the virgin magnetization curves of **5%Dy@1-Y** with **1-Dy** indicate a Dy:Y ratio of about 3:97 in the doped sample (Extended Data Fig. 5a); by contrast inductively coupled plasma mass spectrometry of the same sample gave a Dy:Y ratio of about 7:93; thus, we describe **5%Dy@1-Y** as 5 ± 2 Dy:95 ± 2 Y. The magnetization reversal dynamics of **1-Dy** and **5%Dy@1-Y** were probed at high temperatures by alternating current (a.c.) measurements under zero applied d.c. field, and at low temperatures by d.c. waveform methods[43] (Extended Data Fig. 4, Supplementary Figs. 29–31, and Supplementary Tables 5 and 6). The a.c. and d.c. magnetic susceptibility of **1-Dy** are in close agreement, indicating that all Dy ions present in the sample contribute to the slow magnetization reversal, in accord with bulk sample purity. The magnetization reversal rates ($\tau^{-1}$) for **1-Dy** follow an Arrhenius law above 105 K, where $\tau^{-1} = 10^{-4} \exp(-U_{eff}/k_B T)$ with $U_{eff} = 1,843(11)$ cm⁻¹, $A = -11.55(5) \log_{10}(s)$ (where the lattice attempt time $\tau_0 = 10^4$ and $k_B$ is the Boltzmann constant; Fig. 2b), indicating magnetic reversal by concatenated one-phonon transitions via the Orbach mechanism. The experimentally determined $U_{eff}$ of **1-Dy** is in good agreement with the predicted value (1,809 cm⁻¹), and greater than that for the related complex [Dy{N(Si^iPr₃)₂}₂][Al{OC(CF₃)₃}₄] (642(12) cm⁻¹)[24], the first dysprosocenium complex [Dy(Cp^{ttt})₂][B(C₆F₅)₄] (1,237(28) cm⁻¹)[7,44], and the previous record of

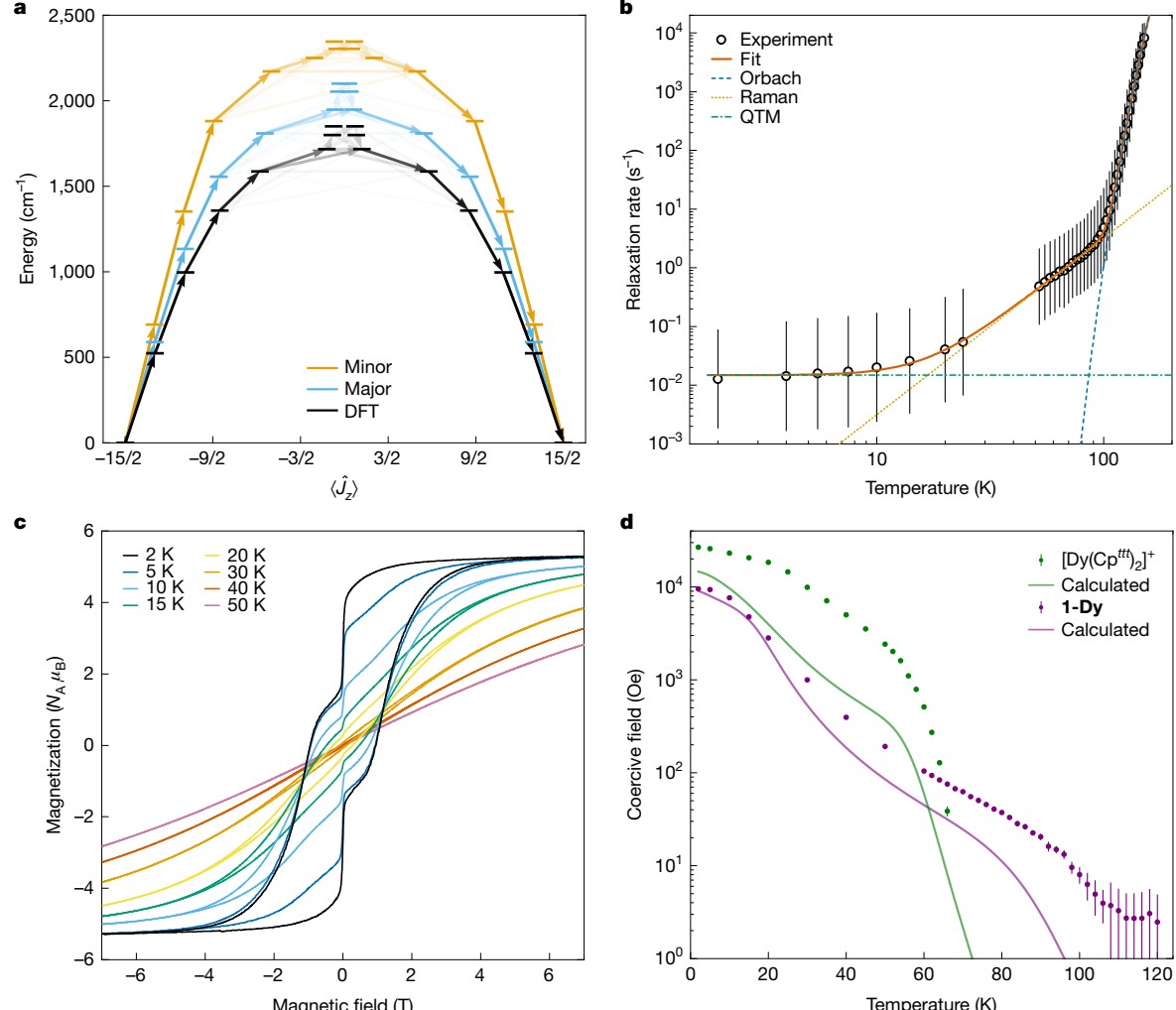

**Fig. 2 | Magnetization reversal behaviour of 1-Dy. a**, Calculated energy barriers to magnetization reversal for models of the cation in **1-Dy**. Crystal field states from CASSCF-SO wavefunction, expressed in terms of the expectation value of their projection of the total angular momentum on the quantization axis, $\langle \hat{J}_z \rangle$. Minor component with N–Dy–N = 165° (orange), major disorder component of the XRD structure of **1-Dy** with N–Dy–N = 150° (blue) and DFT-optimized geometry with N–Dy–N = 146° without point charges (black). The opacity of the arrows represents the relative magnetic-dipole transition probability (averaged over $x$, $y$ and $z$) normalized from each departing state and commencing with unit population in $|-15/2\rangle$; only pathways towards $|+15/2\rangle$ are shown. **b**, Temperature dependence of the magnetization reversal rate; >50 K

from a.c. susceptibility data and <30 K from d.c. waveform data (Supplementary Table 5). The dashed blue line given by $\tau^{-1} = 10^{-A}\exp(-U_{eff}/k_B T)$, the dotted yellow line is given by $\tau^{-1} = 10^R T^n$, the dot-dashed green line given by $\tau^{-1} = 10^{-Q}$, and the solid red line is the sum, with $U_{eff} = 1,843(11)$ cm$^{-1}$, $A = -11.55(5)$ log$_{10}$(s), $R = -5.5(1)$ log$_{10}$(s$^{-1}$ K$^{-n}$), $n = 3.01(6)$ and $Q = 1.83(2)$ log$_{10}$(s). The error bars represent 1 estimated standard deviation of the distribution of rates. **c**, Magnetization hysteresis measured with a sweep rate of 22 Oe s$^{-1}$. **d**, Experimental (points) and calculated (lines) coercive field, $H_C$, versus temperature of **1-Dy** (purple) and [Dy(Cp$^{ttt}$)$_2$][B(C$_6$F$_5$)$_4$] (ref. 44; green). The error bars represent half of the difference between positive and negative field sweeps.

1,687(13) cm$^{-1}$ set by [Dy(OAd)$_2$(18-crown-6)][I$_3$] (ref. 45). Magnetization reversal rates between 10 K and 105 K show a power-law profile $\tau^{-1} = 10^R T^n$ with $R = -5.5(1)$ log$_{10}$(s$^{-1}$ K$^{-n}$) and $n = 3.01(6)$ (where the Raman pre-factor $C = 10^R$ and the Raman temperature exponent is $n$), indicative of a two-phonon Raman scattering process, whereas below 10 K, the rate becomes independent of temperature, indicating quantum tunnelling of the magnetization (QTM) with a rate constant of $10^{1.83(2)}$ s (about 68 s; Fig. 2b). The dilute sample **5%Dy@1-Y** shows comparable reversal rates at high temperature, confirming the molecular origin of these properties, but reaches slower rates at low temperature owing to partial quenching of QTM (Extended Data Fig. 5b,c). At 2 K, the magnetization reversal timescale of **1-Dy** is about 100 s; thus, no 100-s magnetic blocking temperature ($T_{B100s}$) could be determined, whereas $T_{B100s}$ is approximately 16 K for **5%Dy@1-Y**; dysprosium bis-cyclopentadienyl complexes and their derivatives have shown $T_{B100s}$ values up to 72 K (refs. 7–17).

The magnetic hysteresis of **1-Dy** was investigated by SQUID magnetometry with d.c. field sweep rates of 22 Oe s$^{-1}$, with the applied magnetic field accurately calibrated by the palladium reference (Fig. 2c,d, and Extended Data Figs. 3c,d and 6). At low temperatures, we observe large steps at zero field, which are common for monometallic single-molecule magnets (SMMs) and arise from rapid magnetic reversal by QTM, consistent with the a.c. data[43]. At 2 K, we find 50% remanent magnetization ($M_R$) at zero field and a coercive field $H_C = 9.5$ kOe (similar results are also observed for the dilute sample **5%Dy@1-Y**; Extended Data Fig. 5b,d); $H_C$ is far smaller than that for [(C$_5^i$Pr$_5$)Dy(μ-I)$_3$Dy(C$_5^i$Pr$_5$)], which has $H_C > 140$ kOe below 60 K, arising from strong intramolecular exchange coupling from its mixed-valent electronic structure[13]. However, whereas magnetic hysteresis is rapidly closing for [Dy(Cp$^{ttt}$)$_2$][B(C$_6$F$_5$)$_4$] at 60 K ($H_C = 511$ Oe at 60 K, dropping to $H_C = 39$ Oe at 66 K, that is, closing at a rate of about 80 Oe K$^{-1}$)[44], $H_C$ for **1-Dy** drops slowly (about 2 Oe K$^{-1}$) and remains non-zero until 100 K,

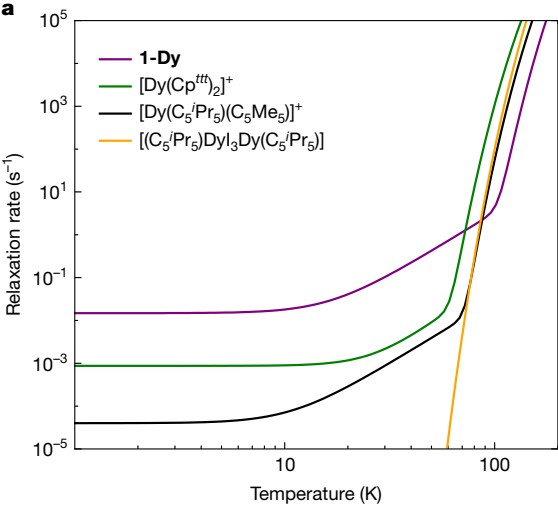

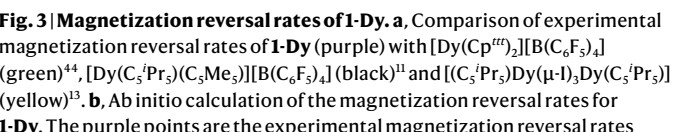

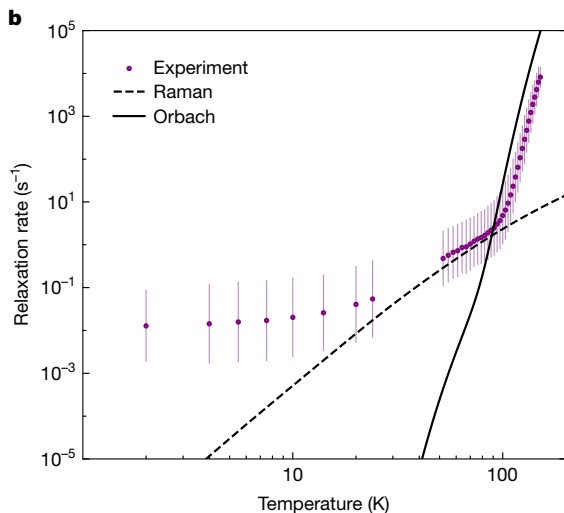

**Fig. 3 | Magnetization reversal rates of 1-Dy. a**, Comparison of experimental magnetization reversal rates of **1-Dy** (purple) with $[Dy(Cp^{ttt})_2][B(C_6F_5)_4]$ (green)[44], $[Dy(C_5{}^iPr_5)(C_5Me_5)][B(C_6F_5)_4]$ (black)[11] and $[(C_5{}^iPr_5)Dy(\mu\text{-I})_3Dy(C_5{}^iPr_5)]$ (yellow)[13]. **b**, Ab initio calculation of the magnetization reversal rates for **1-Dy**. The purple points are the experimental magnetization reversal rates

for **1-Dy** (Supplementary Table 5). The error bars represent 1 estimated standard deviation of the distribution of rates. The solid black line is the ab initio-calculated Orbach rate and the dashed black line is the ab initio-calculated Raman rate.

above which $H_C$ and $M_R$ reach a plateau and the hysteresis is closed within error (Fig. 2d and Extended Data Fig. 3c). The small coercive fields and remanent magnetizations for **1-Dy** at high temperatures (for example, 37 Oe and 0.001 $N_A$ $\mu_B$ at 80 K, where $N_A$ is the Avogadro constant and $\mu_B$ is the Bohr magneton) gives hysteresis loops that can be described as 'soft' (Fig. 2d and Supplementary Fig. 32). As SMMs are superparamagnets, their hysteresis behaviour derives from the underlying molecular spin dynamics[46]. Comparison of the magnetization reversal rates of **1-Dy** with $[Dy(Cp^{ttt})_2][B(C_6F_5)_4]$ (ref. 7), $[Dy(C_5{}^iPr_5)(C_5Me_5)][B(C_6F_5)_4]$ (ref. 11) and $[(C_5{}^iPr_5)Dy(\mu\text{-I})_3Dy(C_5{}^iPr_5)]$ (ref. 13; Fig. 3a) shows that although **1-Dy** has faster spin dynamics than any of these other complexes below 70 K, magnetization reversal does not switch from the Raman regime to the Orbach regime until 105 K ($\tau_{switch}$ = 6.7 s)[47]; above 90 K, its spin dynamics are up to 100 times slower than those of the other complexes, leading to the observed slow closing of the hysteresis.

To probe the origin of the magnetic reversal mechanisms in **1-Dy**, we performed ab initio spin dynamics calculations using our recently developed methods[48,49], which involves: (1) optimization of the crystal structure and calculation of phonon modes with periodic DFT; (2) calculation of the embedded molecular electronic structure and spin-phonon coupling with CASSCF-SO; and (3) simulations of one- and two-phonon spin dynamics with a semi-classical master equation (Extended Data Fig. 7, Supplementary Figs. 42–46 and Supplementary Table 8). The optimized crystal structure is a reasonable match with the experimental structure, but with a reduced N–Dy–N angle of 146° compared with the XRD geometry (150.1(5)° and 165.3(8)°), slightly diminishing the calculated energy barrier to magnetization reversal (Fig. 2a). The calculated phonon dispersion and density of states (DOS; Supplementary Figs. 42 and 43) show several small off-Γ imaginary modes (<20i cm$^{-1}$) that were not computationally feasible to remove owing to the large size of the primitive unit cell; however, their presence does not impact the results (Supplementary Fig. 45). Calculation of the spin-phonon coupling and magnetic reversal rates gives excellent agreement with experiment (Fig. 3b), confirming the molecular origin of the large $U_{eff}$ value. They also confirm faster Raman rates for **1-Dy** than previously observed[7,44] and calculated[50] for $[Dy(Cp^{ttt})_2][B(C_6F_5)_4]$; at 50 K (where Raman dominates in both complexes), the calculated reversal rate of **1-Dy** is nearly 10 times faster than that for $[Dy(Cp^{ttt})_2]$ $[B(C_6F_5)_4]$. These faster two-phonon dynamics are driven by the

much larger spectral density (phonon DOS weighted by spin-phonon coupling strength) at low energy in **1-Dy** than $[Dy(Cp^{ttt})_2][B(C_6F_5)_4]$ (refs. 44,50,51; Extended Data Fig. 7).

To confirm the origin of the high temperature coercivity and soft hysteresis behaviour of **1-Dy**, we extended our spin dynamics calculations to non-zero magnetic fields, allowing us to directly simulate the magnetic hysteresis from first principles calculations of spin-phonon coupling. Owing to the large magnetic anisotropy, we must compute the spin dynamics as a function of both field strength and orientation, which we then use to propagate state populations and simulate the hysteresis experiment for both **1-Dy** and $[Dy(Cp^{ttt})_2][B(C_6F_5)_4]$ (ref. 50; Extended Data Figs. 8 and 9, Supplementary Figs. 47–49 and Supplementary Table 8). We underpredict $H_C$ in both cases, but reproduce the features that distinguish the two complexes (Fig. 2d). This confirms our experiments showing that hysteresis closes slowly for **1-Dy** so that it maintains non-zero $H_C$ up to 100 K, despite $[Dy(Cp^{ttt})_2]$ $[B(C_6F_5)_4]$ having larger $H_C$ at low temperature[50]. The simulations also confirm that suppression of the Orbach mechanism until 105 K in **1-Dy** is the origin of the observed behaviour, and additionally highlight the competition between one- and two-phonon mechanisms in the field dependence of the spin dynamics (Extended Data Fig. 8d) and hence determination of $H_C$ (Extended Data Fig. 9c).

To conclude, we have shown that enhancing magnetic anisotropy in **1-Dy** allows open magnetic hysteresis to persist in a molecule up to 100 K, and one can envisage that suppression of two-phonon Raman rates by using rigid ligands that reduce the low-energy spectral density[51] may lead to slower molecular spin dynamics across the whole temperature range. This represents a change in regime where the limits imposed by the magnetic anisotropy of dysprosium cyclopentadienyl SMMs can be overcome. We also identify that there is still scope to increase magnetic anisotropy with more linear E–Dy–E angles (that is, >150°; E = monodentate donor atom) and more charge-dense ligands (for example, dianions). If these features can be combined, this may then deliver magnetic memory at even higher temperatures than seen for **1-Dy**, providing multiple pathways for future exploration; we note that axial dysprosium complexes of the general formula $[Dy(Cp^R)(OAr)][B(C_6F_5)_4]$ ($Cp^R = C_5R_5$; Ar = aryl) have previously been proposed as target SMMs that combine the rigidity of $Cp^R$ rings with the stronger electrostatic interactions provided by monodentate ligands[52].

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

## Methods

### General synthetic procedures

All manipulations were conducted under argon with the strict exclusion of oxygen and water using standard Schlenk and glove box techniques. Glassware was flame-dried under vacuum before use. Argon was passed through a column of activated 3-Å molecular sieves and copper catalyst before use. $C_6D_6$ was purchased anhydrous, degassed and stored under argon over activated 3-Å molecular sieves. $C_6H_6$ and $n$-hexane were refluxed over molten potassium for 3 days, distilled and stored under argon over a potassium mirror. $C_6H_5F$ and hexamethyldisiloxane (HMDSO) were refluxed over $CaH_2$ for 3 days, distilled and stored under argon over activated 3-Å molecular sieves. $LnI_3$ (ref. 53), $[K\{N(Si^iPr_3)_2\}]$ (ref. 24) and $[HNEt_3][Al\{OC(CF_3)_3\}_4]$ (ref. 12) were synthesized by literature methods, whereas **1-Y** and **2-Y** were prepared by procedures analogous to those for **1-Dy** and **2-Dy**, respectively.

### Synthesis of 1-Dy

A mixture of **2-Dy** and $HN\{Si^iPr_3\}_2$, (1.620 g), prepared as described below from $DyI_3$ (2.720 g, 5.00 mmol) and $[K\{N\{Si^iPr_3\}_2\}]$ (1.660 g, 4.50 mmol), was treated with $[HNEt_3][Al\{OC(CF_3)_3\}_4]$ (1.07 g, 1.0 mmol) in $C_6H_6$ (20 ml) for 18 h at 40 °C. All volatiles were removed in vacuo, and the residues washed with $n$-hexane (about 3× 5 ml). The residues were dissolved in $C_6H_5F$ (about 2 ml) and layered under $n$-hexane (about 20 ml), affording on diffusion a pale-yellow oil beneath colourless crystals of $[HNEt_3][Al\{OC(CF_3)_3\}_4]$. The oil was decanted into a clean flask and the crystallization process repeated nine times, at which point no additional $[HNEt_3][Al\{OC(CF_3)_3\}_4]$ was observed in the residual oil by $^1H$ NMR spectroscopy. The final supernatant was decanted and **1-Dy** was recrystallized by slow evaporation of residual solvent at ambient pressure. The crystalline material was triturated with excess $n$-hexane and dried in vacuo. Yield: 0.236 g, 0.132 mmol, 13% taking $[HNEt_3]$ $[Al\{OC(CF_3)_3\}_4]$ as the limiting reagent. $^1H$ NMR (400.07 MHz, $C_6H_5F$): $\delta$ 3.26 (s), 2.69 (s), 2.59 (s), 2.33 (s), 1.95 (s), 1.67 (s), 1.21 (s). $^{19}F$ NMR (376.40 MHz, $C_6H_5F$): $\delta$ −92.95 (s, $CF_3$). Anal. calcd. for $C_{52}H_{82}AlDyF_{36}$ $N_2O_4Si_4$ (1,785.00 g mol⁻¹): C, 34.99; H, 4.63; N, 1.57. Found: C, 32.92; H, 4.20; N, 1.53. Fourier transform infrared spectroscopy (attenuated total reflectance (ATR), microcrystalline): $\tilde{v}$ = 2,954 (m), 2,870 (m), 1,461 (m), 1,349 (m), 1,210 (s), 965 (s), 727 (s), 540 (s), 437 (s).

### Synthesis of 2-Dy

A suspension of $DyI_3$ (1.630 g, 3.00 mmol) and $[K\{N(Si^iPr_3)_2\}]$ (3.303 g, 9.00 mmol) in $C_6H_6$ (20 ml) was stirred in a sealed ampoule at 100 °C for 18 h. The volatiles were then removed in vacuo, and the residues extracted into $n$-hexane (about 3× 10 ml). The volatiles were again removed to afford a viscous yellow oil containing a mix of **2-Dy** and $H\{N(Si^iPr_3)_2\}$; **2-Dy** was subsequently isolated by crystallization from a concentrated HMDSO solution at −35 °C, followed by washing with cold HMDSO (<5 ml, −35 °C). Yield: 0.417 g, 0.510 mmol, 17%. $^1H$ NMR (400.07 MHz, $C_6D_6$): $\delta$ 1.79 (2), 0.98 (s). 0.70 (s), −0.45 (s), −2.38 (s), −9.14 (s). Anal. calcd. for $C_{36}H_{81}DyN_2Si_4$ (816.90 g mol⁻¹): C, 52.93; H, 9.99; N, 3.43. Found: C, 52.80; H, 10.34; N, 3.37. Fourier transform infrared spectroscopy (ATR, microcrystalline): $\tilde{v}$ = 2,938 (s), 2,857 (s), 2,757 (w), 2,705 (w), 1,515 (w), 1,461 (s), 938 (s), 880 (s), 688 (s), 652 (s).

### NMR spectra

NMR spectra were recorded at 298 K on a Bruker AVIII HD 400 cryoprobe spectrometer operating at 400.07 MHz ($^1H$), 100.61 MHz ($^{13}C$), 376.40 MHz ($^{19}F$) or 79.48 MHz ($^{29}Si$) MHz. Chemical shifts are reported in ppm and coupling constants in Hz. $^1H$ and $^{13}C\{^1H\}$ DEPTQ NMR spectra, where DEPTQ is the distortionless enhancement by polarization transfer including the detection of quaternary nuclei pulse sequence, recorded in $C_6D_6$ are referenced to the solvent signal[54]. NMR spectra recorded in $C_6H_5F$ were locked to an internal sealed capillary of $C_6D_6$, with $^1H$ NMR spectra referenced using the highest intensity peak of the lower-frequency fluoroarene multiplet ($\delta_H$ 6.865) and $^{13}C\{^1H\}$ DEPTQ spectra referenced to $C_6D_6$. $^{19}F$ ($C_7H_5F_3/CDCl_3$) and $^{29}Si\{^1H\}$ DEPT90 (SiMe$_4$) spectra were referenced to external standards. Paramagnetic **1-Dy** and **2-Dy** did not exhibit resonances in their $^{13}C\{^1H\}$ DEPTQ and $^{29}Si\{^1H\}$ DEPT90 NMR spectra, and we were not able to assign their $^1H$ NMR spectra; resonances between +400 ppm and −400 ppm are noted.

### Infrared spectra

ATR infrared spectra of microcrystalline powders were recorded using a Bruker Alpha Fourier transform infrared spectrometer with a platinum-ATR module at ambient temperature. Elemental analysis (C, H, N) and inductively coupled plasma mass spectrometry samples were carried out by M. Jennings and A. Davies at the Microanalytical Service, Department of Chemistry, the University of Manchester. Elemental analysis values obtained for **1-Ln** and **2-Ln** typically gave carbon compositions that were lower than expected values. This phenomenon has commonly been ascribed to incomplete combustion owing to carbide formation in air- and moisture-sensitive complexes, as we have previously observed reproducibly for lanthanide $\{N(Si^iPr_3)_2\}$ complexes[21,24,55–57]. We also note that inconsistent results have been highlighted as an underlying issue with this analytical technique[58], with elemental analyses of fluorine-rich complexes such as **1-Ln** highlighted as being particularly problematic[59].

### Single-crystal X-ray diffraction

Single crystals of all compounds were mounted in Fomblin YR-1800 oil and XRD data were collected on a Rigaku FR-X diffractometer equipped with a HyPix-6000HE photon-counting pixel array detector and a mirror-monochromated X-ray source using Cu K$\alpha$ radiation (wavelength $\lambda$ = 1.5418 Å). Intensities were integrated from data recorded on 0.5° frames by $\omega$-axis rotation, which is the axis perpendicular to the incident X-ray beam. Cell parameters were refined from the observed positions of all strong reflections in each data set. A Gaussian grid face-indexed with a beam profile was applied for all structures[60]. The structures were solved using SHELXT[61]; the datasets were refined by full-matrix least-squares on all unique $F^2$ values, where $F$ is the crystallographic structure factor[61]. Anisotropic displacement parameters were used for all non-hydrogen atoms with constrained riding hydrogen geometries, with the exception of borohydride hydrogen atoms, which were located in the difference map and refined isotropically; the hydrogen atom isotropic displacement parameter ($U_{iso}$) was set at 1.2 (1.5 for methyl groups) times the equivalent isotropic displacement parameter ($U_{eq}$) of the parent atom. The largest features in final difference syntheses were close to heavy atoms and were of no chemical relevance. CrysAlisPro[60] was used for control and integration, and SHELX[61,62] was employed through OLEX2[63] for structure solution and refinement. ORTEP-3[64] and POV-Ray[65] were used for molecular graphics. Plots of electron density maps were generated on Mercury 4.0[66].

### Powder X-ray diffraction

Microcrystalline samples of **1-Dy**, **1-Y**, and **5%Dy@1-Y** were mounted in Fomblin YR-1800 oil and powder XRD data were collected at 100 K between an incident angle ($\theta$) of 3° and 70°, with a detector distance of 150 mm and a beam divergence of 1.0 mRad (ref. 67), using a Rigaku FR-X rotating anode single-crystal X-ray diffractometer with Cu K$\alpha$ radiation ($\lambda$ = 1.5418 Å) with a Hypix-6000HE detector and an Oxford Cryosystems nitrogen flow gas system. The instrument was calibrated using the collected data, with the instrument model refined using diffraction peak positions measured at multiple detector angles. The data were collected, reduced and integrated using CrysAlisPro software[60]. Peak hunting and unit cell indexing was performed using TOPAS software[27]. Le Bail profile analysis was performed using JANA2006 software[67]. The two broad peaks centred around approximately 16° and 42° 2$\theta$ are owing to scatter from the Fomblin YR-1800 oil.

## Magnetic measurements

Magnetic measurements were performed using a Quantum Design MPMS3 SQUID magnetometer. Samples of **1-Dy** (20.7 mg) and **5%Dy@1-Y** (30.8 mg) were crushed with a mortar and pestle under an inert atmosphere, then loaded into a borosilicate glass NMR tube with eicosane flakes (**1-Dy** 21.3 mg; **5%Dy@1-Y** 27.0 mg). Samples were gently heated to melt the eicosane and then cooled. The tube was flame-sealed (about 3 cm) under dynamic vacuum and mounted in a straw using Kapton tape. Data were corrected for the diamagnetism of the straw, NMR tube and eicosane using calibrated blanks, for the shape of the sample using Quantum Design Geometry Simulator (factors 0.996–1.034), and for the diamagnetism of the sample (estimated as the molecular weight (g mol$^{-1}$) multiplied by $-0.5 \times 10^{-6}$ cm$^3$ K mol$^{-1}$). The data for **5%Dy@1-Y** were processed assuming 3.46% Dy, calibrated using the magnetization saturation value of 5.20 $N_A \mu_B$ for **1-Dy** at 2 K, 7 T (Extended Data Fig. 5a). To calibrate the magnetic field, measurements were performed on a palladium standard at 298 K under identical field-charging conditions, outlying data points were removed, and the field correction versus the reported field was fitted to a sum of B-splines[68] (24, 21, 41 and 49 knots for 0–7 T, ±3 T, ±5 T and ±7 T) in Mathematica 12.3[69].

Direct-current susceptibility measurements were performed on **1-Dy** with a 0.1-T field between 300 K and 2 K with a constant sweep rate of 0.5 K min$^{-1}$. For zero-field cooled, virgin magnetization, a.c. and waveform measurements, a magnetic reset was performed before cooling the sample. Susceptibility and hysteresis measurements were performed in vibrating sample magnetometer (VSM) mode with a 5 mm vibration amplitude and 2 s averaging time, except for the 2 K and 5 K hysteresis and virgin magnetization for which a 0.5 mm amplitude was used to minimize vibrational heating. The latter data were noisy: outliers (with large errors) were removed and data with $|H| > 0.2$ T were smoothed using parabolic-weighted adjacent averaging. Hysteresis measurements were performed with a constant sweep rate of 22 Oe s$^{-1}$ between ±7 T at 2–50 K, and ±3 T at 60–120 K for **1-Dy** and between ±7 T at 2 K, ±5 T at 10–20 K and ±3 T at 30–50 K for **5%Dy@1-Y**. The coercive field and remanent magnetization were determined by interpolating the $x$ and $y$ intercept, respectively; values are reported as the average from positive and negative sweeps, with the uncertainty defined as half the difference.

Alternating-current susceptibility measurements were performed at 55–151 K (**1-Dy**) or 55–131 K (**5%Dy@1-Y**). Measurements were performed using 8 frequencies per decade between 0.1 Hz and 1,000 Hz (55–124 K) or between 1 Hz and 1000 Hz (127–151 K) for **1-Dy** and 4 frequencies per decade between 0.1 Hz and 647 Hz (55–127 K) or between 1 Hz and 647 Hz (131 K) for **5%Dy@1-Y**. An oscillating field of 5 Oe was used for 0.1–563 Hz, and a 2 Oe oscillating field for 750 Hz and 1,000 Hz. Averages for **1-Dy** were performed for 4 s or 20 cycles (0.1–10 Hz), and 2 s or 10 cycles (13–1,000 Hz), whichever was longer. For **5%Dy@1-Y** and $T \geq 91$ K, averages were performed for 10 s or 50 cycles (0.1–10 Hz), 4 s or 20 cycles (13–87 Hz), or 2 s or 10 cycles (114–647 Hz). For **5%Dy@1-Y** and $T \leq 87$ K, averages were performed for 20 s or for 100 cycles (0.1–87 Hz) or 10 s or for 50 cycles (114–647 Hz). Waveform measurements were performed below 24 K with a field of ±8 Oe, a field sweep rate of 700 Oe s$^{-1}$, a fixed moment range of unity, and VSM mode with an amplitude of 1 mm (0.5 mm for 2 K) and a 0.5 s averaging time[43]. Waveform frequencies in mHz (number of square-wave periods) for **1-Dy** (* indicates frequencies used for **5%Dy@1-Y**): 0.1 (2)*, 0.32 (2), 0.56 (2)*, 1.0 (2), 1.8 (2)*, 3.1 (2), 5.5 (3)*, 9.9 (4), 13 (5), 17 (6), 21 (6)*, 28 (7), 36 (8)*, 46 (9), 57 (10)*. In- and out-of-phase susceptibilities were extracted in CC-FIT2[70,71], disabling filtering based on error values and using a field window of ±0.3 Oe to discard data points from before and after the measurement. Alternating-current and waveform data were fit to the generalized Debye model, and the temperature dependence of magnetic reversal rates was fitted in CC-FIT2[70,71].

## Electronic structure

CASSCF-SO calculations on **1-Dy** and **2-Dy** were performed with OpenMolcas 23.02[72]. The XRD geometry was used for each disorder component, excluding the anion for **1-Dy**, as well as using the optimized geometry of the cation of **1-Dy** from periodic DFT. We used the second-order Douglas–Kroll–Hess relativistic Hamiltonian[73], ANO-RCC basis sets[74] (VTZP for Dy, VDZP for N and coordinated allyl/alkene C, VDZ all other atoms), and the resolution of the identity approximation of two electron integrals with the Cholesky 'atomic compact' auxiliary basis set[75]. State-averaged CASSCF calculations were performed with a 9 electrons in 7 4$f$ orbital active space, considering 21 roots for total spin $S = 5/2$, 224 roots for $S = 3/2$ and 490 roots for $S = 1/2$. The CASSCF states were mixed with spin–orbit coupling including 21 $S = 5/2$ states, 128 $S = 3/2$ states and 130 $S = 1/2$ states. We projected the $^6H_{15/2}$ multiplet from the spin-orbit states to obtain the composition of the low-lying states in the $m_J$ basis using molcas_suite[76].

## Spin dynamics calculations

Following our established methodology[48,49,77,78], the solid-state structure of **1-Dy** was optimized with DFT using the program VASP 6.1.2[79–82] with the PBE (Perdew–Burke–Ernzerhof) functional[83]. We note that the choice of functional has a direct bearing on the calculated phonon DOS, which in turn directly impacts magnetization reversal rates; there is yet to be a systematic study of the impact of such considerations for SMMs, and we are currently working on this as a standalone study. We used a plane-wave-basis set up to 900 eV (determined via convergence testing) and sampled the electronic structure at the Γ point. Atomic positions and cell shape were optimized to a force tolerance of 0.001 eV Å$^{-1}$ starting from the XRD data of the major component. Phonons were calculated with phonopy[84].

To obtain the spin-phonon coupling, CASSCF-SO calculations were performed where the crystalline environment around a single **1-Dy** cation was represented by a spherical cluster of unit cells (40 Å radius) composed of point charges (obtained from gas-phase DFT calculations on the cationic and anionic components of **1-Dy** using CHELPG[85]), and then surrounded further by a spherical conductor (Kirkwood solvent model with dielectric constant $\varepsilon \to \infty$), which screens the unphysical surface charges to reproduce the Madelung potential[48]. We used a 9-in-7 active space for 18 $S = 5/2$ states only, and other details as described above. The spin–phonon coupling for each phonon (index $j$) at each q-point $\partial \hat{H} / \partial Z_{q,j}$ was evaluated using our linear vibronic coupling method without recourse to a model Hamiltonian[77,78].

Magnetization reversal rates were calculated with Tau[86], considering one-phonon (Orbach and direct) and two-phonon (Raman-I) rates using perturbation theory expressions (equations 40, 41 and 46–49 in ref. 49) with a magnetic field of 2 Oe along the main anisotropy axis. Integration was performed over anti-Lorentzian phonon lineshapes (equation 11 in ref. 49, using full-width at half-maximum linewidths $\Gamma = 0.1$–100 cm$^{-1}$) within an equivalent range of $\mu \pm 2\sigma$ (95%) using the trapezoidal method with 40 equidistant steps, and restricted to $\omega < 496.7$ cm$^{-1}$ for the two-phonon terms. Very little dependence on linewidth is observed (Supplementary Fig. 46), much less than the distributions of experimental rates; $\Gamma = 10$ cm$^{-1}$ was chosen as the best compromise. Q-point meshes from $1 \times 1 \times 1$ to $3 \times 3 \times 3$ gave indistinguishable rates; meshes other than $1 \times 1 \times 1$ included several imaginary phonons modes: removing them or setting the frequency to its absolute value result in near-identical rates. The spectral density was calculated as the product of the phonon DOS and the spin-phonon coupling strength per mode[87].

## Hysteresis modelling

One- and two-phonon rates were calculated with Tau[86] as a function of field magnitude (at 2 Oe, and every 0.1 T from 0.1 T to 7 T), orientation (50 points with a hemispherical Fibonacci lattice[88]), and temperature

(every 2 K from 2 K to 120 K). (Note 1: our approach differs from Soncini and co-workers as we calculate phonons, spin–phonon coupling, and spin dynamics ab initio, whereas they used a model for spin–phonon coupling and assumed a Debye-like phonon spectrum[89,90]. Note 2: the one-phonon rates show that, at low temperature, the direct mechanism within the ground doublet is quickly turned on in small fields and has a power-law field dependence at higher fields[91], whereas the Orbach mechanism dominates with minimal field dependence at higher temperatures; the two-phonon Raman-I rates have a weak field dependence[92], but it is non-zero owing to the splitting of the ground doublet interacting with a changing cross-section of phonons that can mediate the scattering process.) Similar calculations were performed for [Dy(Cp$^{ttt}$)$_2$][B(C$_6$F$_5$)$_4$] using phonons from ref. 50, and restricting $\hbar\omega < 99$ cm$^{-1}$ for two-phonon rates. QTM rates are not included in our modelling, noting that the closing of the hysteresis at high temperature is dominated by phonon-driven processes where QTM is not relevant.

To calculate the hysteresis curve, we linearly interpolate state energies ($E_i$) and magnetic moments ($M_i$) as a function of field, and interpolate $\log_{10}(\tau^{-1})$ with two-dimensional cubic splines as a function of field and temperature. Then, we use symmetry relations to obtain values for negative fields (states $n$ and $\bar{n}$ are Kramers pairs):

$$\tau^{-1}(H) = \tau^{-1}(-H)$$

$$E_n(H) = E_{\bar{n}}(-H)$$

$$M_n(H) = -M_{\bar{n}}(-H)$$

Initial-state populations (at −7 T or −3 T) were set to Boltzmann equilibrium. The magnetic field was swept at a rate of $s = 22$ Oe s$^{-1}$ towards either +7 T or +3 T, and state populations were propagated in time with a time step of $\Delta t = 1$ ms (required to converge calculated coercive fields to within 1 Oe; Supplementary Fig. 49). At time step $t$, the population vector $P_t$ is:

$$P_t = (P_{t-1} - P_t^{eq})e^{-\tau_t^{-1}\Delta t} + P_t^{eq}$$

where $P_t^{eq}$ is the equilibrium population at $t$, $P_{t-1}$ is the population at the previous time step, and $\tau_t^{-1}$ is the calculated magnetization reversal rate at the given field strength, orientation and temperature. Only populations of the lowest four states were considered owing to the large energy gaps to excited states. At time step $t$, the net magnetization $M_t$ is:

$$M_t = -\sum_n M_{n,t}P_{n,t}$$

where $M_{n,t}$ is the magnetic moment of state $n$ and $P_{n,t}$ is its population. $M_t$ is converted into $M(H)$ with $H_t = H_0 - t \times \Delta t \times s$, and the reverse sweep obtained by inverting the forwards sweep around both field and magnetization axes. This was performed for each field orientation and the resulting loops were integrated over the hemispherical grid to give the powder data. The coercive field was obtained by interpolation of powder data for each isotherm as a function of magnetic field.

## Gas-phase DFT

A gas-phase DFT geometry optimization was performed on **1-Y**. Calculations were performed with the hybrid PBE0 functional[93], with the def2-TZVP basis on all atoms[94], and the D4 dispersion correction[95], in ORCA 5.0.2[96]. The geometry optimization was started from the crystal structure geometry. Quantum theory of atoms in molecules analysis was performed with Critic2[97,98].

## Data availability

Single-crystal X-ray data are available free of charge from the Cambridge Crystallographic Data Centre, reference numbers 2370040–2370044.

A preprint of this work was deposited on ChemRxiv on 15 July 2024 at https://doi.org/10.26434/chemrxiv-2024-36vjp. Research data files supporting this publication are available from figshare at https://doi.org/10.6084/m9.figshare.26262656. All other datasets generated and analysed during the current study are available from the corresponding authors upon reasonable request. Source data are provided with this paper.

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

**Acknowledgements** We thank the European Research Council (StG-851504 and CoG-816268), the Leverhulme Trust (RPG-2023-025), and the UK EPSRC (EP/R002605X/1, EP/P001386/1, EP/S033181/1 and EP/T011289/1) for funding, and the EPSRC UK National Electron Paramagnetic Resonance Service for access to the SQUID magnetometer. Computational resources and services were provided by the University of Manchester Computational Shared Facility, the National Computational Infrastructure (NCI), and the Pawsey Supercomputing Research Centre's Setonix Supercomputer (https://doi.org/10.48569/18sb-8s43), with funding from the Australian Government and the Government of Western Australia.

**Author contributions** J.E.-K. and D.P.M. provided the original synthetic concept. J.E.-K. synthesized and characterized the compounds, collected, solved and refined the single-crystal XRD datasets, and collected the powder XRD datasets. G.F.S.W. performed further XRD refinement, finalized the single-crystal XRD data, and solved and refined the powder XRD datasets. G.K.G. and W.J.A.B. collected and interpreted the magnetic data. B.E.A. and G.K.G. performed the ab initio calculations. B.E.A. and N.F.C. developed the code. N.F.C. supervised the magnetism and calculation components and D.P.M. supervised the synthetic component. N.F.C. and D.P.M. directed the research and wrote the paper, with contributions from all authors.

**Competing interests** The authors declare no competing interests.

**Additional information**
**Correspondence and requests for materials** should be addressed to Nicholas F. Chilton or David P. Mills.

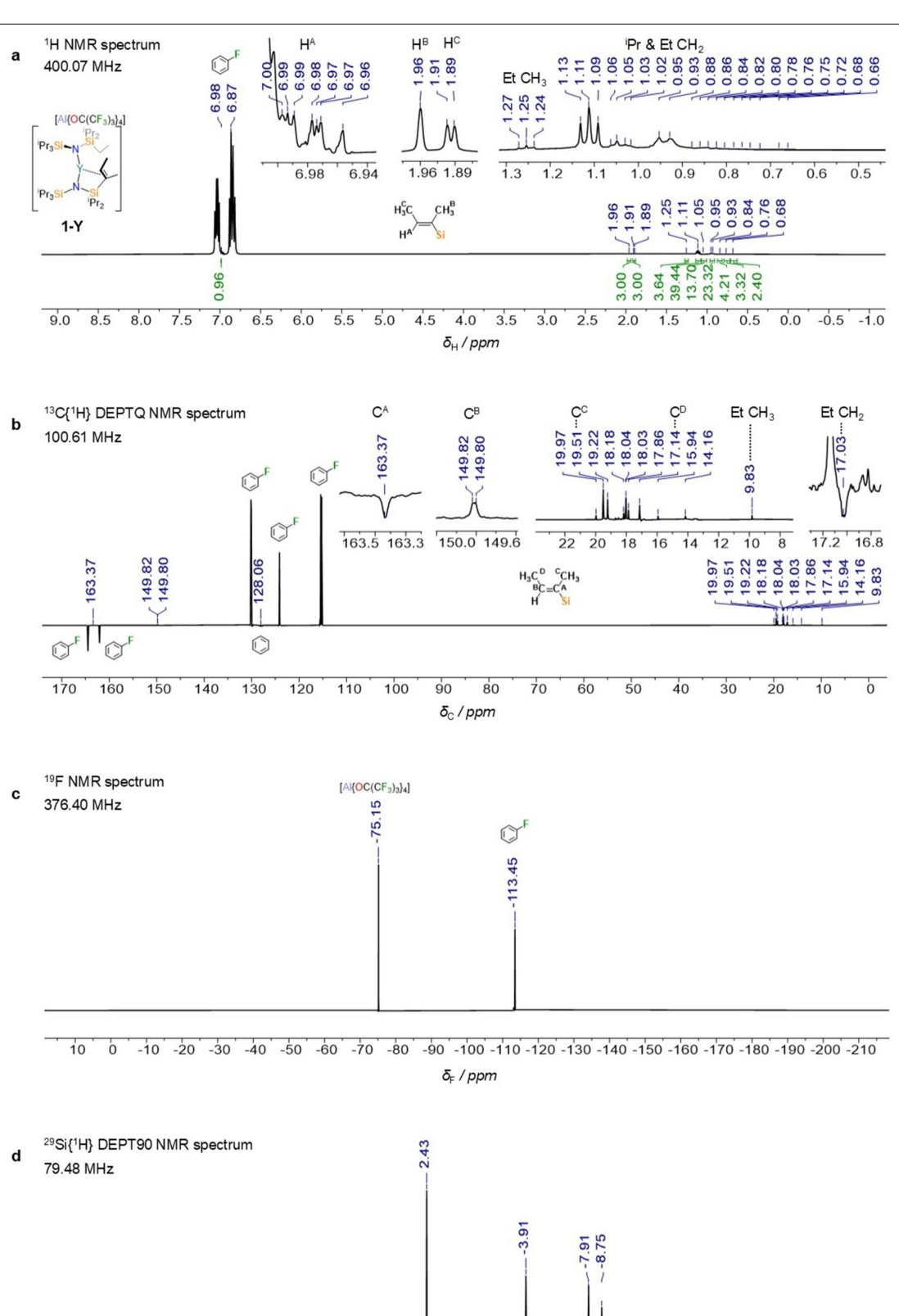

**Extended Data Fig. 1 | Annotated NMR spectra of 1-Y in C$_6$H$_5$F solution at 298 K. a.** $^1$H NMR spectrum (400.07 MHz). **b.** $^{13}$C{$^1$H} DEPTQ NMR spectrum (100.61 MHz). **c.** $^{19}$F NMR spectrum (376.40 MHz). **d.** $^{29}$Si{$^1$H} DEPT90 NMR spectrum (79.48 MHz).

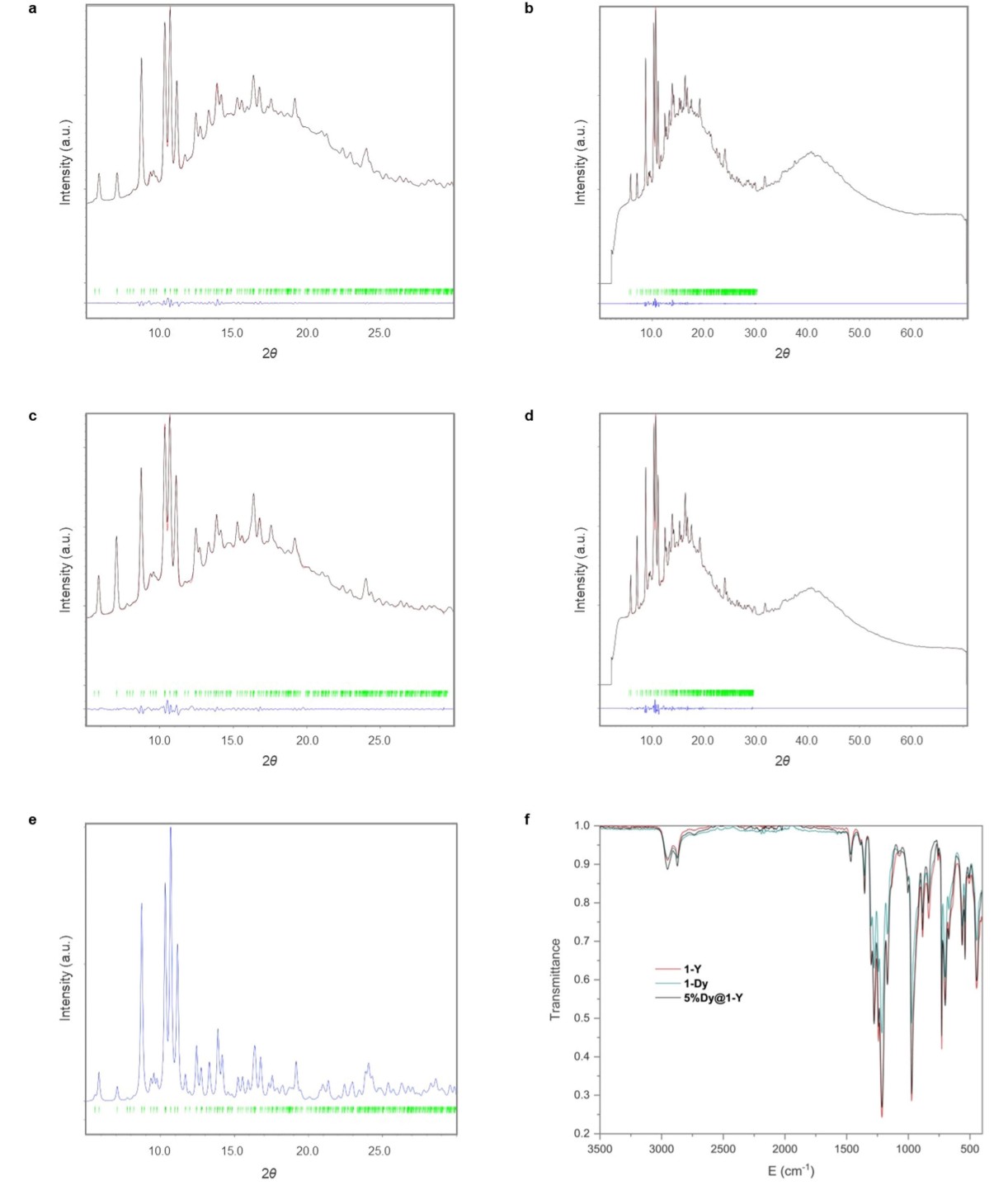

**Extended Data Fig. 2 | Selected solid-state characterisation data for 1-Ln.**
**a**. Powder XRD pattern of **1-Dy** in the range used for Le Bail profile fitting. **b**. Full powder XRD pattern of **1-Dy**. **c**. Powder XRD pattern of **5%Dy@1-Y** in the range used for Le Bail profile fitting. **d**. Full powder XRD pattern of **5%Dy@1-Y**. **e**. Calculated powder XRD pattern for **1-Dy** based on single crystal XRD data, FWHM = 0.2. **f**. Overlaid FT-IR (ATR) spectra of microcrystalline **1-Dy**, **1-Y** and **5%Dy@1-Y**. For **a-e**, black = observed data, red = calculated profile, green = reflection positions, blue = observed-calculated residuals.

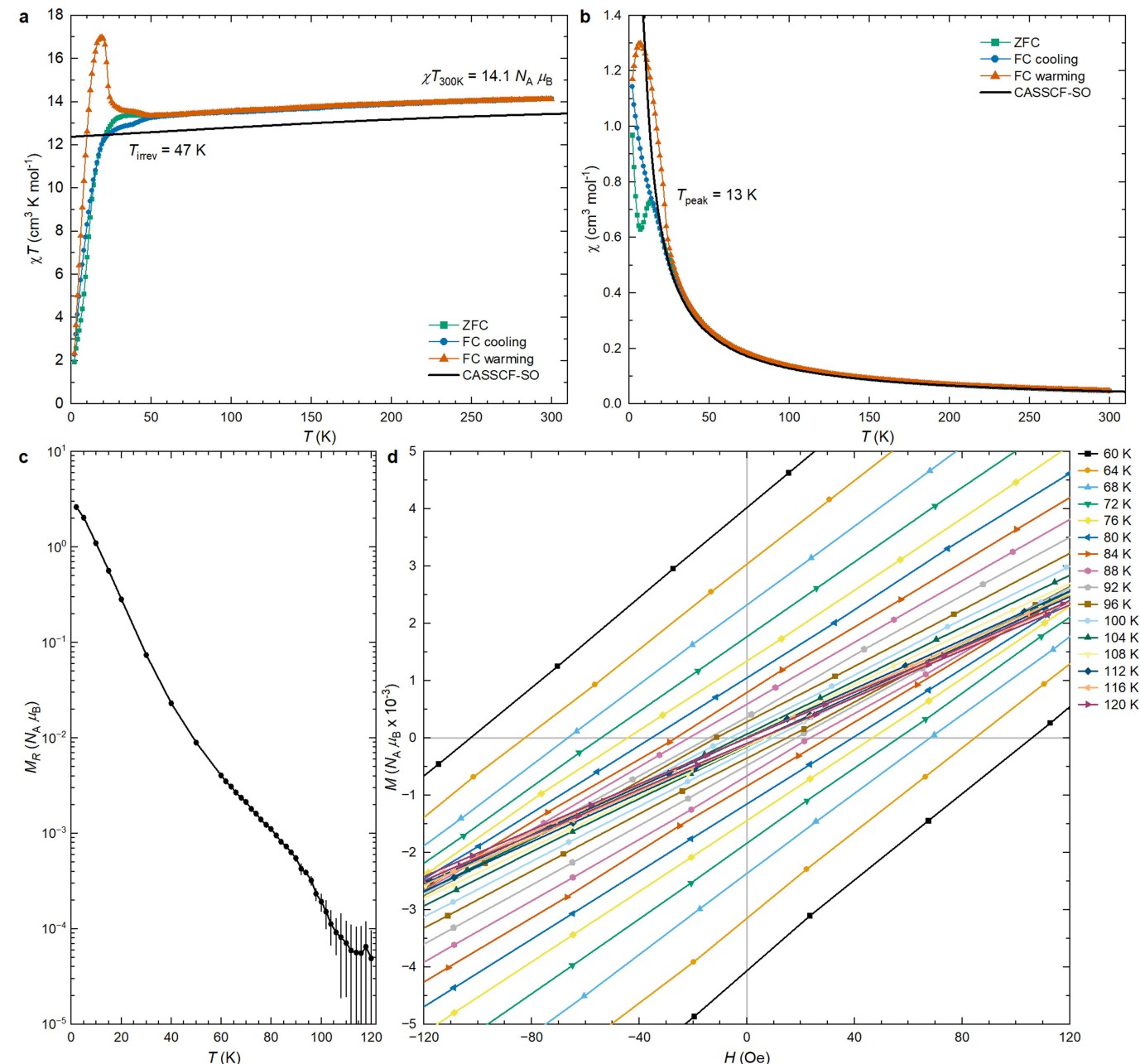

**Extended Data Fig. 3 | Selected d.c. magnetic data for 1-Dy. a**. Magnetic susceptibility temperature product ($\chi T$) vs. temperature measured under a 0.1 T applied d.c. magnetic field. **b**. Magnetic susceptibility ($\chi$) vs. temperature measured under a 0.1 T applied d.c. magnetic field on warming after cooling in zero-field (ZFC), on cooling in field (FC cooling) and on warming in field (FC warming), along with the CASSCF-SO calculated equilibrium trace. Sweep rate is 0.5 K min$^{-1}$. **c**. Temperature dependence of the remnant magnetisation ($M_R$). Error bars indicate half the difference between positive and negative sweeps. **d**. Hysteresis loops from 60 to 120 K in 4 K increments and fields swept from −3 T to +3 T, zoomed in to ±120 Oe. Sweep rate is 22 Oe s$^{-1}$.

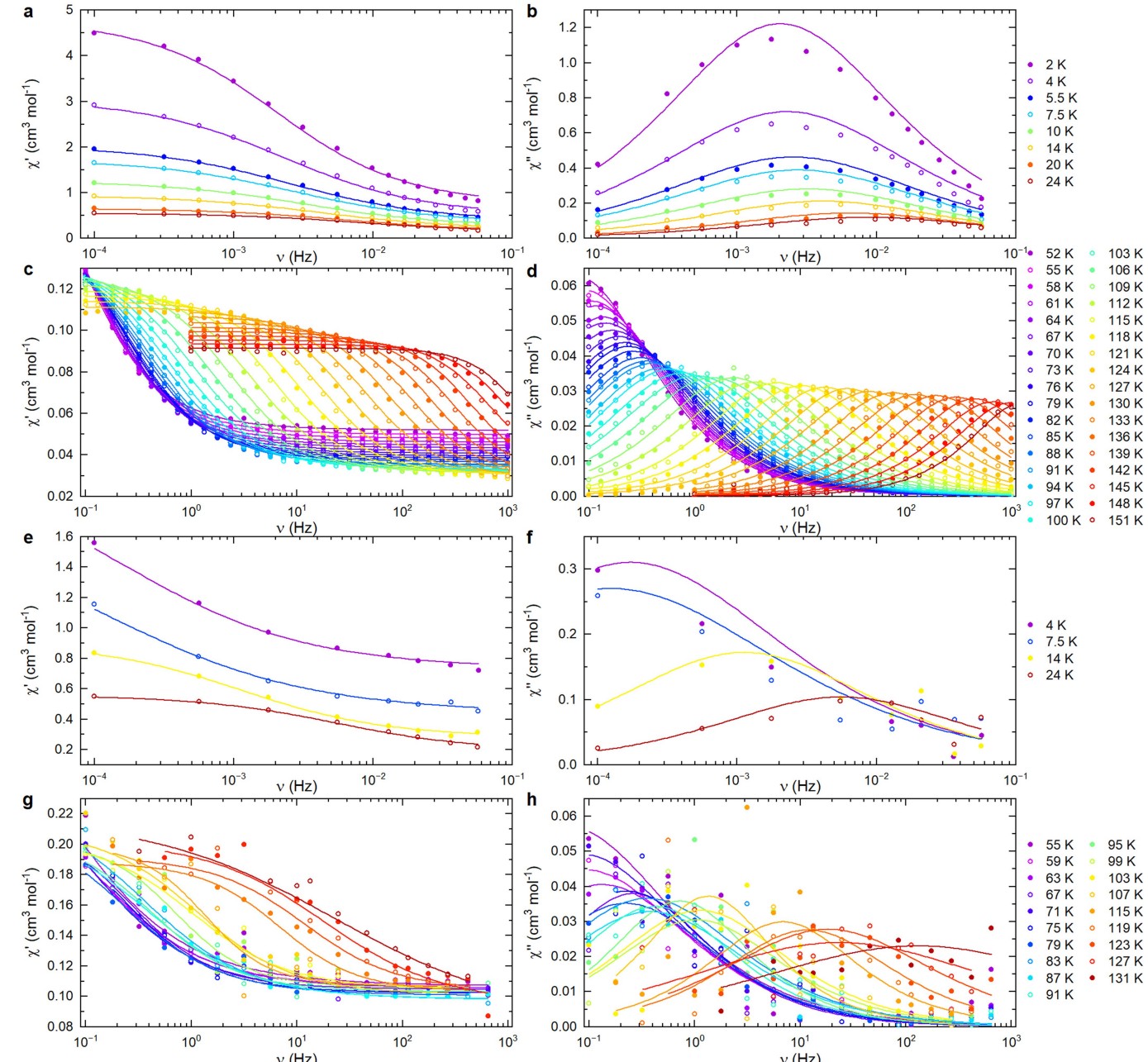

**Extended Data Fig. 4 | Frequency-dependent in-phase (χ′, left) and out-of-phase (χ″, right) field susceptibility for 1-Dy and 5%Dy@1-Y. a,b.** Waveform data for **1-Dy. c,d.** a.c. data for **1-Dy. e,f.** Waveform data for **5%Dy@1-Y.**

**g,h.** a.c. data for **5%Dy@1-Y.** All data fitted in zero d.c. field, using the generalised Debye model in CCFIT-2.

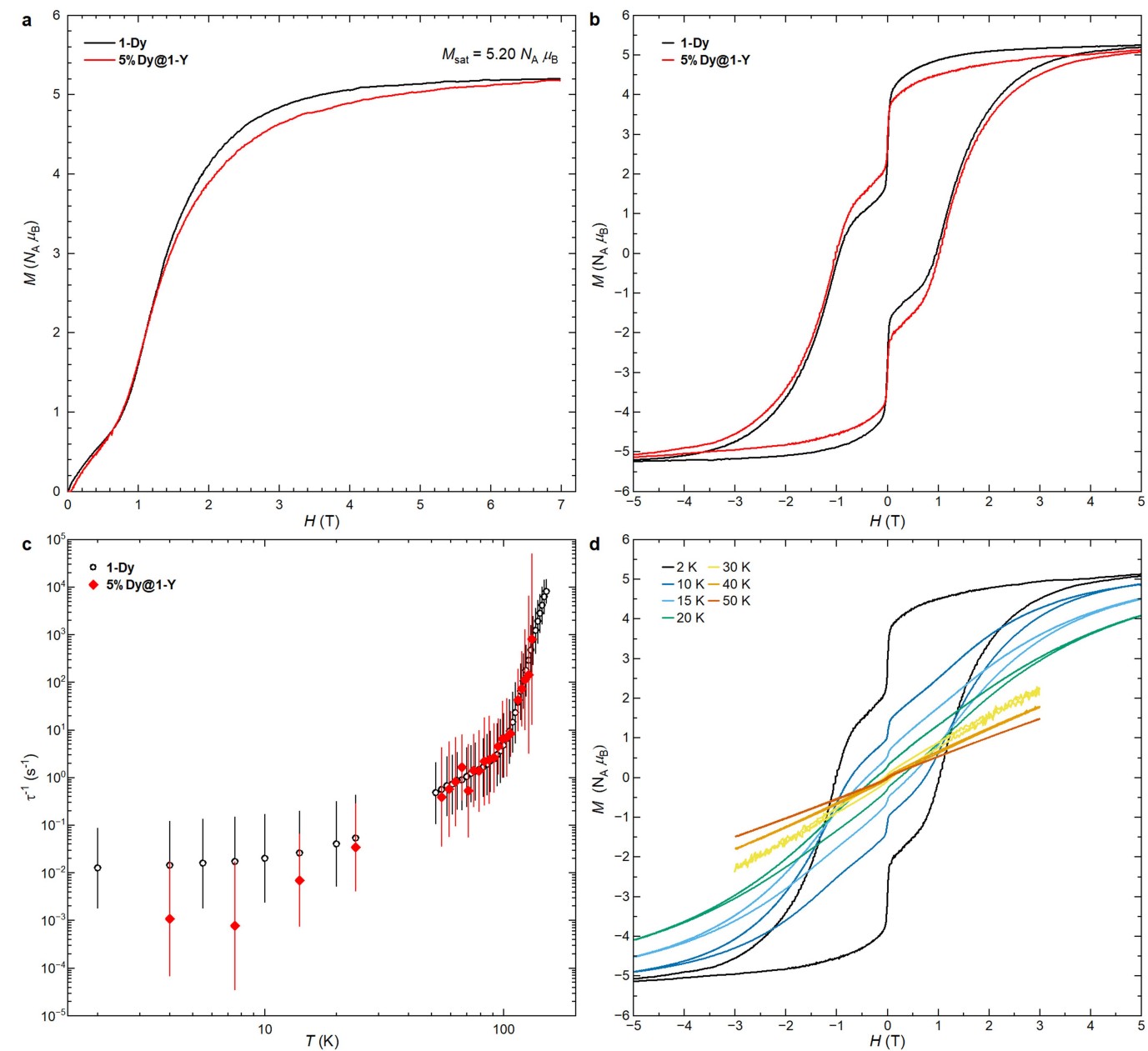

**Extended Data Fig. 5 | Comparison of magnetic data of 1-Dy with 5%Dy@1-Y.**
**a**. Virgin magnetisation curves for **1-Dy** and **5%Dy@1-Y** at 2 K used to calibrate the doping level in **5%Dy@1-Y**. **b**. Hysteresis curves at **1-Dy** and **5%Dy@1-Y** at 2 K. Sweep rate is 22 Oe s⁻¹, and hysteresis is performed between +7 and –7 T.

**c**. Temperature-dependence of the relaxation rates for **1-Dy** and **5%Dy@1-Y**. Error bars represent 1 estimated standard deviation of the distribution of rates. **d**. Hysteresis curves between 2 and 50 K for **5%Dy@1-Y** with a sweep rate of 22 Oe s⁻¹.

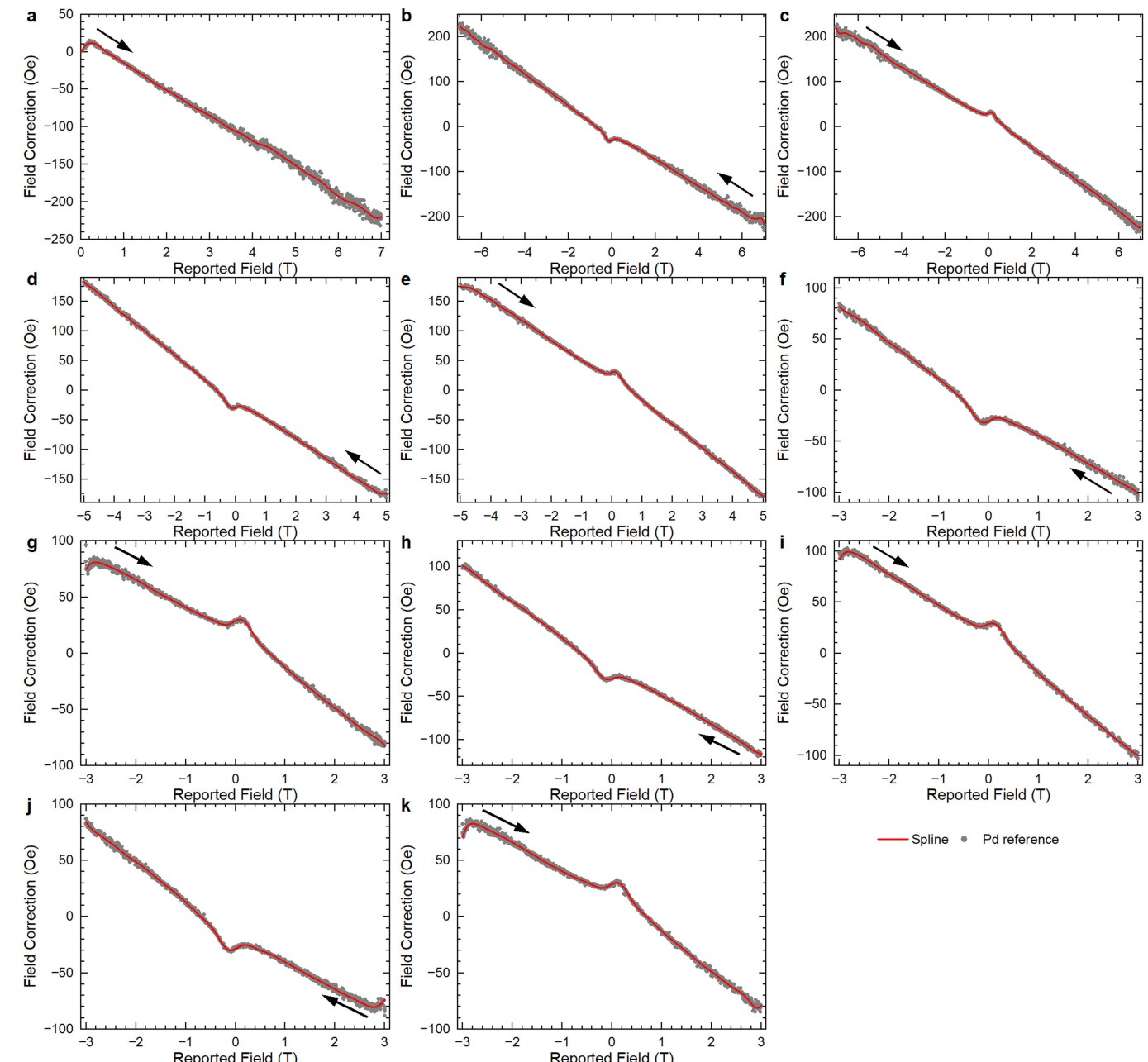

**Extended Data Fig. 6 | Calibration of the magnetic field for d.c. field-swept measurements using a Pd reference sample. a**. 0 → +7 T curve after magnet reset. **b,c**. ±7 T hysteresis after previous sweep. **d,e**. ±5 T hysteresis after previous ±5 T hysteresis. **f,g**. ±3 T hysteresis after approaching +3 T from +7 T at 700 Oe s⁻¹. **h,i**. ±3 T hysteresis after approaching +3 T from +5 T at 700 Oe s⁻¹. **j,k**. ±3 T hysteresis after previous ±3 T hysteresis. Temperature is 298 K, sweep rate is 22 Oe s⁻¹; arrows show sweep direction.

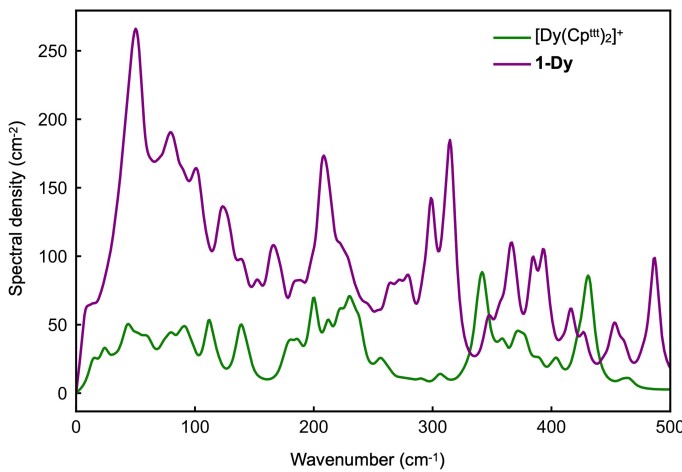

**Extended Data Fig. 7 | Phonon spectral density of [Dy(Cp$^{ttt}$)$_2$][B(C$_6$F$_5$)$_4$] (green) and 1-Dy (purple).** Modes are broadened by anti-Lorentzian functions with a full-width half maximum linewidth of 10 cm$^{-1}$. The spectral density is the product of the phonon DoS and the spin-phonon coupling strength per mode. Phonons sampled on a 1×1×1 q-mesh, as used in rate simulations.

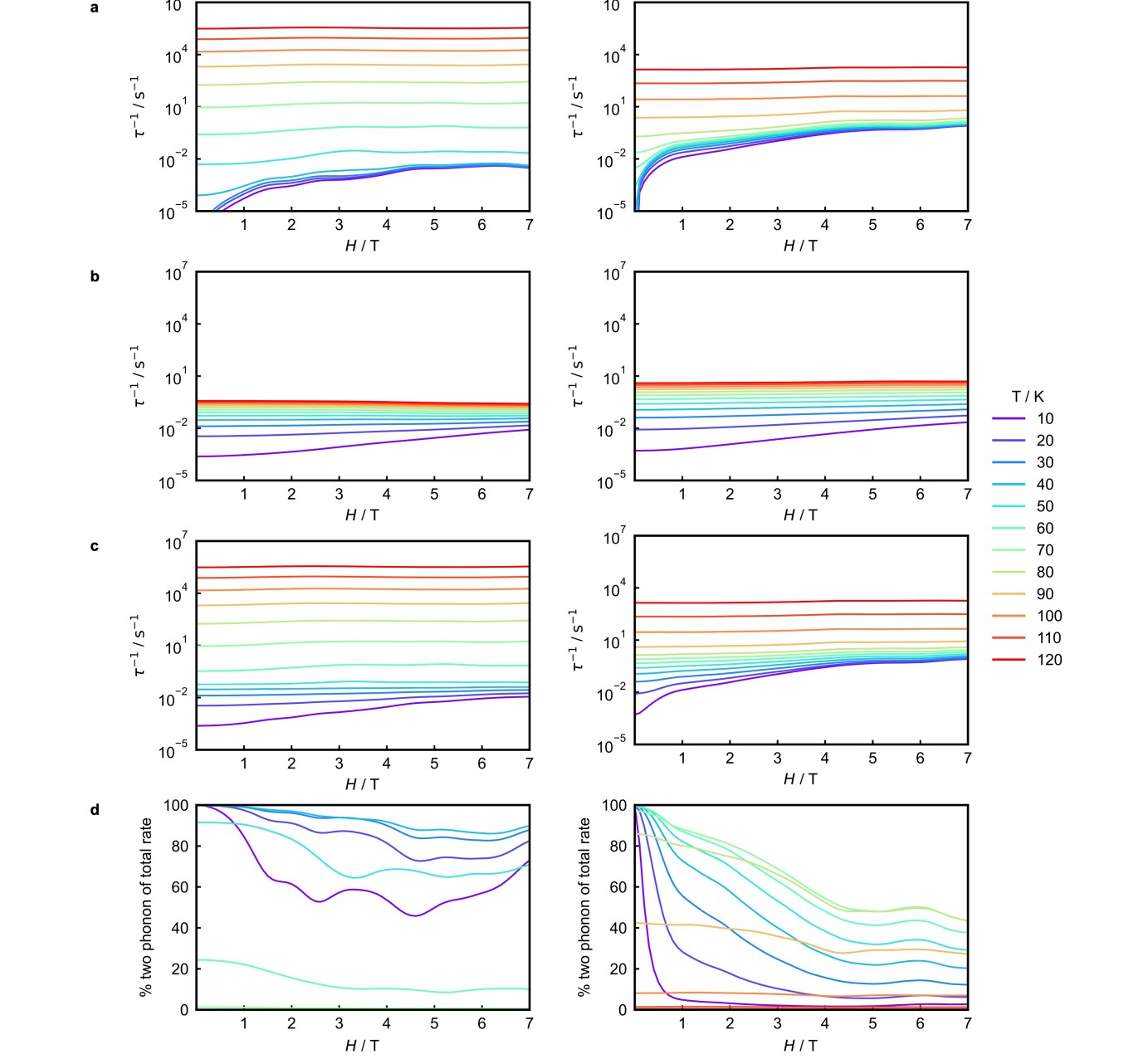

**Extended Data Fig. 8 | Calculated magnetic reversal rates for [Dy(Cp$^{ttt}$)$_2$] [B(C$_6$F$_5$)$_4$] (left) and 1-Dy (right) as a function of magnetic field, aligned along the principal magnetic axis. a**. One-phonon reversal rate. **b**. Two-phonon reversal rate. **c**. Total reversal rate. **d**. Percentage that the two-phonon Raman-I rates have of the total rate; note that for [Dy(Cp$^{ttt}$)$_2$][B(C$_6$F$_5$)$_4$] all percentages for T ≥ 70 K are *ca*. 0 %.

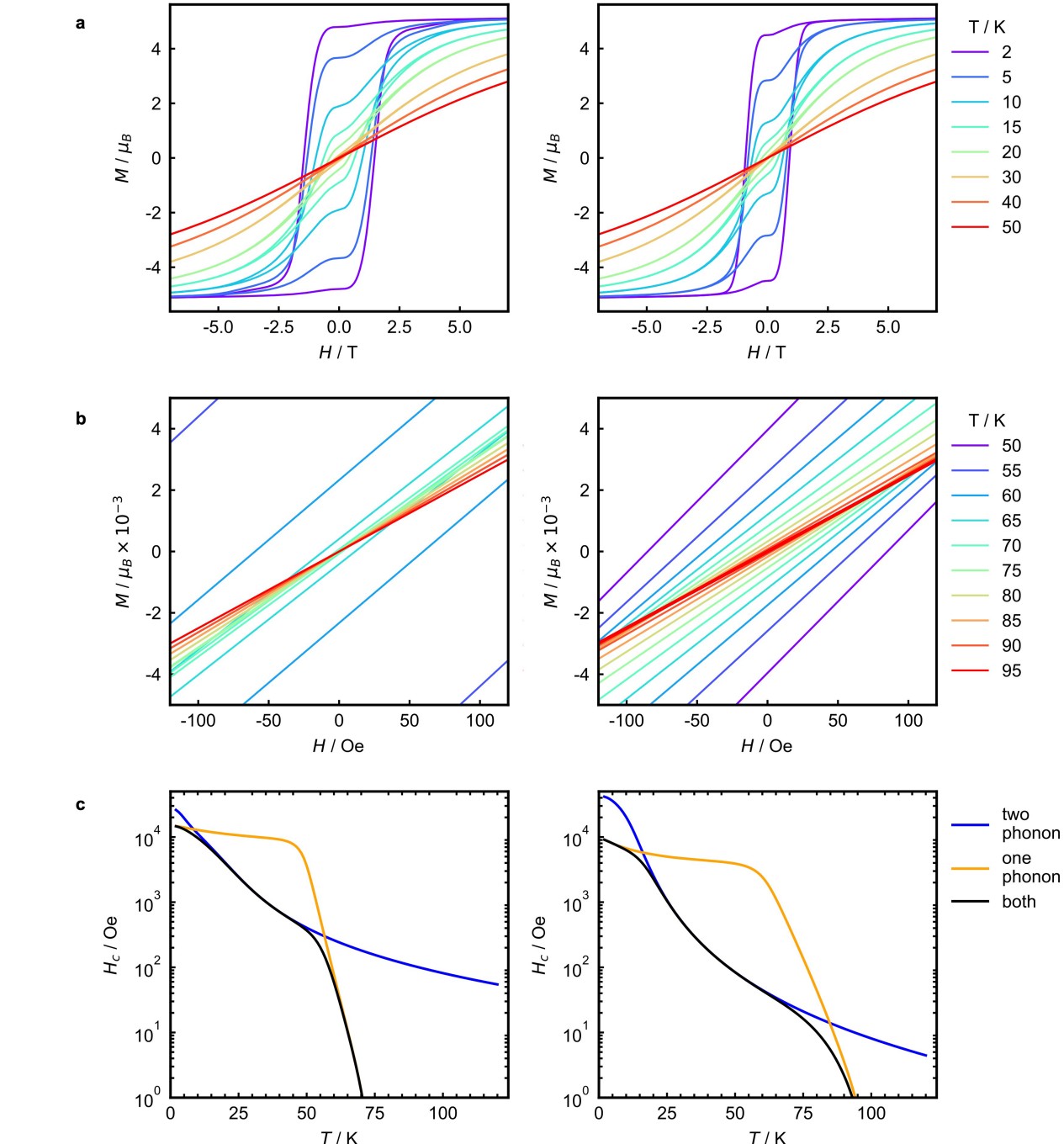

**Extended Data Fig. 9 | *Ab initio*-calculated magnetic hysteresis loops and magnetic relaxation rates of 1-Dy (left) and [Dy(Cp$^{ttt}$)$_2$][B(C$_6$F$_5$)$_4$] (right).** **a**. Calculated magnetic hysteresis loops with a sweep rate of 22 Oe s$^{-1}$ and timestep of 1 ms, simulated between 7 and −7 T. **b**. Zoom in of calculated magnetic hysteresis loops with a sweep rate of 22 Oe s$^{-1}$ and timestep of 1 ms, simulated between 3 and −3 T. **c**. Calculated coercive field, $H_C$, vs. temperature, simulated including only two phonon rates (blue), one phonon rates (orange), and both rates (black). Coercive field calculated with a timestep of 100 ms.

**Extended Data Table 1 | CASSCF-SO-calculated electronic structure of 1-Dy[a]**

| Energy (cm⁻¹) | Energy (K) | $g_x$ | $g_y$ | $g_z$ | Angle (deg) | $\langle J_z \rangle$ | Wavefunction |
|---|---|---|---|---|---|---|---|
| **Major Crystal Structure Geometry** | | | | | | | |
| 0.00 | 0.00 | 0.00 | 0.00 | 19.89 | 0.00 | ±7.49 | 94.92% \|±15/2> |
| 588.88 | 841.39 | 0.00 | 0.00 | 16.93 | 0.70 | ±6.48 | 92.44% \|±13/2> |
| 1133.73 | 1619.86 | 0.02 | 0.03 | 14.03 | 2.10 | ±5.45 | 89.84% \|±11/2> |
| 1555.71 | 2222.78 | 0.27 | 0.29 | 11.36 | 14.32 | ±4.29 | 82.61% \|±9/2> + 6.20% \|±5/2> |
| 1808.96 | 2584.62 | 0.55 | 1.02 | 9.56 | 40.04 | ±2.72 | 55.97% \|±7/2> + 16.90% \|±3/2> + 5.73% \|±5/2> + 5.14% \|±9/2> + 5.06% \|±1/2> |
| 1948.18 | 2783.54 | 3.78 | 6.71 | 7.94 | 80.25 | ±0.31 | 18.67% \|∓1/2> + 14.94% \|±7/2> + 14.42% \|±1/2> + 14.26% \|∓5/2> + 13.88% \|±5/2> + 8.58% \|∓7/2> |
| 2053.54 | 2934.08 | 1.33 | 2.86 | 14.95 | 88.21 | ±0.19 | 28.72% \|±5/2> + 26.98% \|∓3/2> + 12.17% \|∓5/2> + 7.51% \|±1/2> + 6.20% \|±3/2> + 5.79% \|∓7/2> |
| 2100.17 | 3000.71 | 0.36 | 0.68 | 19.41 | 88.92 | ±0.14 | 24.00% \|∓1/2> + 21.88% \|±3/2> + 18.12% \|±1/2> + 13.45% \|∓3/2> + 7.34% \|∓5/2> + 5.13% \|±5/2> |
| **Minor Crystal Structure Geometry** | | | | | | | |
| 0.00 | 0.00 | 0.00 | 0.00 | 19.90 | 0.00 | ±7.50 | 95.03% \|±15/2> |
| 691.38 | 987.84 | 0.00 | 0.00 | 16.91 | 1.06 | ±6.49 | 92.09% \|±13/2> |
| 1352.74 | 1932.78 | 0.01 | 0.01 | 14.00 | 1.25 | ±5.49 | 90.11% \|±11/2> |
| 1880.92 | 2687.44 | 0.08 | 0.09 | 11.35 | 5.96 | ±4.46 | 89.76% \|±9/2> |
| 2171.60 | 3102.76 | 2.32 | 2.60 | 10.64 | 45.99 | ±2.50 | 66.48% \|±7/2> + 9.42% \|±5/2> + 7.51% \|±3/2> |
| 2250.08 | 3214.89 | 0.53 | 4.32 | 12.38 | 75.45 | ±1.02 | 43.14% \|±1/2> + 17.85% \|±3/2> + 12.10% \|±7/2> + 10.03% \|±5/2> |
| 2303.70 | 3291.50 | 0.88 | 4.86 | 9.23 | 86.58 | ±0.24 | 23.45% \|±5/2> + 20.95% \|±1/2> + 19.65% \|∓3/2> + 7.69% \|∓1/2> + 7.64% \|±3/2> + 7.27% \|∓5/2> |
| 2345.87 | 3351.76 | 1.20 | 6.78 | 13.58 | 83.27 | ±0.34 | 39.32% \|±5/2> + 26.06% \|±3/2> + 10.62% \|∓1/2> + 10.59% \|∓3/2> |
| **DFT Optimised Structure** | | | | | | | |
| 0.00 | 0.00 | 0.00 | 0.00 | 19.89 | 0.00 | ±7.49 | 94.83% \|±15/2> |
| 523.63 | 748.16 | 0.00 | 0.00 | 16.95 | 0.39 | ±6.48 | 92.63% \|±13/2> |
| 996.52 | 1423.81 | 0.04 | 0.05 | 14.07 | 3.79 | ±5.43 | 89.64% \|±11/2> |
| 1358.02 | 1940.32 | 0.42 | 0.43 | 11.39 | 15.06 | ±4.26 | 79.55% \|±9/2> + 6.77% \|±5/2> |
| 1586.69 | 2267.04 | 0.35 | 1.06 | 9.09 | 32.87 | ±2.89 | 57.45% \|±7/2> + 15.03% \|±3/2> + 6.94% \|±9/2> + 5.31% \|±5/2> |
| 1717.45 | 2453.88 | 4.12 | 6.16 | 7.52 | 79.02 | ±0.60 | 22.85% \|±5/2> + 18.51% \|±7/2> + 17.77% \|±1/2> + 12.43% \|∓1/2> + 9.72% \|∓5/2> |
| 1799.48 | 2571.08 | 1.32 | 2.89 | 15.81 | 86.21 | ±0.41 | 35.25% \|±3/2> + 21.74% \|∓5/2> + 17.20% \|∓1/2> + 11.83% \|±5/2> |
| 1850.45 | 2643.91 | 0.24 | 0.52 | 19.38 | 87.12 | ±0.37 | 22.57% \|±3/2> + 21.41% \|∓1/2> + 17.53% \|±1/2> + 11.32% \|∓3/2> + 8.57% \|∓5/2> + 6.75% \|±5/2> |

[a]Each row corresponds to a Kramers doublet and gives the energy, effective g-values in the principal, x, y and z directions ($g_x$, $g_y$ and $g_z$), the angle between the $g_z$ value of the excited Kramers doublet and the ground Kramers doublet, the expectation value of the total angular momentum along the z-direction of the ground Kramers doublet ($\langle \hat{J}_z \rangle$) and the wavefunction composition. All data is directly from the CASSCF-SO calculation, except for the Wavefunction, which is necessarily obtained by projection onto a model space of the $^6H_{15/2}$ term.