## [Peer Review File · Nature]

Soft magnetic hysteresis in a dysprosium amide-alkene complex up to 100 K

Corresponding Author: Professor David Mills

Version 0:

Reviewer comments:

Referee #1

(Remarks to the Author)

The manuscript reports on the new record magnetic anisotropy of easy axis type in a Dysprosium complex with almost linear coordination and, consequently, the record barrier for the reversal of the magnetization as a Single-Molecule Magnet.

The work highlights how the current understanding of the correlation between molecular structure and magnetic properties can guide the synthetic design to increase the performances of these materials.

The work is solid, comprehensive, and clearly presented. The authors have made the choice to leave the investigation of the mechanism of reaction to a separate publication. This choice is probably fine if Nature is the journal were the manuscript will be published but should be reconsidered if the manuscript is published in a more specialist journal.

Concerning the significance, I have found that the title is misleading, and, in my opinion, it should be changed. It is not clear what the authors want to communicate with 'soft hysteresis'. The two terms are somehow in conflict, as for a magnet the hysteresis is associated with the 'hardness' of the material. Thus, I was expecting some counterintuitive and completely new mechanism or phenomenon connected to hysteresis. Most likely the authors refer to the almost negligible coercive field at high temperature. However, no efforts were devoted to simulating the hysteresis. Thus, it is not granted that the property emphasized in the manuscript ($H_c=5$ Oe at 102 K, Figure 2D) is fully rationalized, in particular the flattening that is observed above 100 K in Figure 2D.

This new SMM exhibits the record barrier for magnetization reversal, but the under-barrier mechanisms (Raman or QTM) are more efficient than in other reported high-temperature SMMs. As a result, the relaxation time levels at low temperatures to a value that is even lower than the 100 s considered in magnetism textbooks as the threshold for magnetization blocking (see Figure 3).

The authors are crystal clear about the fact that they have succeeded in improving one parameter, magnetic anisotropy, by switching from sandwich complexes to linear ones, at the expense of the others, i.e., the efficiency of low-energy phonons and QTM.

As a result, the manuscript is borderline for acceptance in one of the most generalistic journals. On the contrary, the take-home message on chemical design is very strong.

I have also some technical comments:

- Please, clearly state in the main text how "stable" is the material and how it must be handled.
- Synthesis of 5%Dy@1-Y: Is really 2-Dy mixed with 1-Y, or is it just a typo?
- X-ray powder diffraction: The authors should say if the broad peaks come from the sample holder or from an amorphous phase of the compounds
- Figure S57. The data fitting at the lowest temperature is very poor, but the alpha parameter and its error are reasonable.
- When discussing the results of ab initio calculations the authors refer to Table S6. However, in the SI there are 4 of these tables with slightly different values among them and with the values reported in the main. A comment about the different models used to obtain Tables S6-S9 and the criteria employed to assess which one is the most accurate should be given in the main text.

Referee #2

(Remarks to the Author)

This work cannot be published in any journal due to the poor experimental quality of the reported compounds.

1. The proof of purity for all new compounds is highly questionable.
The carbon values of the elemental analyses for all compounds are:

1-Y: expected 36.49, found 33.27.
1-Dy: expected 34.99, found 32.92.
5%Dy@1-Y: expected 36.42, found 33.47,
2-Y 58.17, found 55.74.
2-Dy 52.93, found 52.80.

It is obvious that the elemental analyses are far off from the expected values. In fact, discrepancies in C values (calc vs found) are 3.22% (1-Y), 3.52% (1-Dy), 2.95% (5%Dy@1-Y), 2.43% 2-Y, 0.13% 2-Dy.

2. It is evident from the moderate R values of all reported compounds, that the quality of the crystals is poor and, in addition, the molecular structures are highly disordered.

1-Y: R1 = 6.64, wR2 = 19.74.
1-Dy: R1 = 9.03, wR2 = 29.19.
5%Dy@1-Y: R1 = 9.38, wR2 = 30.36.
2-Y: R1 = 7.18, wR2 = 17.92.
2-Dy: R1 = 10.70, wR2 = 25.56.

Having said this and keeping in mind that magnetism is extremely sensitive to composition (impurity), any claim for a magneto-structural relationship is obsolete.

3. Another concern refers to the discussion of the metal coordination features of the compounds which is quite confusing: the coordination mode of the metal centers is clearly NOT linear. Apparently, the compounds are diamido pi olefin-metal complexes by no means linear but containing four-coordinate metal centers. Any reasonable textbook (e.g., Cotton-Wilkinson) will tell and falsify this inappropriate and artificial description.

4. There are many self-citations, and other significant references are not mentioned.

Again, the insufficient quality of the central experimental data and the confusing structural discussion disqualifies the paper to be published in any scientific journal.

Referee #3

(Remarks to the Author)

In the field of single-molecule magnets (SMMs), constructing linear two-coordinate trivalent lanthanide (Ln) complexes has long been a dream but is a formidable challenge from the perspective of chemical synthesis due to the large ionic radius and variable coordination pattern of Ln(III) ions. In general, the monodentate ligand with bulky or even super-bulky side groups can serve as an ideal platform to achieve such goal owing to the electrostatic interaction between Ln(III) and ligand. In this work, Mills et al. reported a dysprosium bis(amide)-alkene complex, 1-Dy, featuring relatively short Dy-N bond lengths and a slightly bent N-Dy-N angle at the axial position, and weak Dy-C interactions in the equatorial plane. Although X-ray crystallography demonstrated that this compound doesn't have the strictly two-coordinated linear arrangement, the weak bonding between lanthanide and proximal atoms in the equatorial plane arising from Dy-Calkene and Agostic-type interactions plays a vital role in stabilizing solid-state structures. More importantly, the transverse crystal field (CF) induced by above weak interactions does not significantly diminish axial CF resulting from the strong Dy-N bonding, which was clearly proven by ab initio calculations and QTAIM analysis of isostructural Y(III) compound. The complex exhibits the largest effective energy barrier and the highest blocking temperature for all known SMMs, i.e. 1843 cm⁻¹ and 100 K. The detailed theoretical calculations, including static and dynamic CF considerations, have also been carried out, elucidating the relaxation mechanism of magnetization in 1-Dy. Without doubt, this work reported a remarkable advance in SMM field, and this is going to be a milestone in this direction. I believe it will appeal to a wide cross-section of the chemistry community, including synthetic coordination and organometallic chemists, theoretical chemists as well as materials chemists, and will be cited by every researcher in SMM field. Therefore, I am delighted to recommend acceptance of the manuscript after addressing the following points:

1. Theoretical calculations show that the SMM performance of [Dy(C5iPr5)(C5Me5)][B(C6F5)4] has reached the upper limit of [Dy(CpR)2]⁺ family. In this work, the authors ascribed the larger effective energy barrier of the dysprosium bis(amide)-alkene complex than that of [Dy(C5iPr5)(C5Me5)][B(C6F5)4] to the shorter Dy-N distance and the more dense negative charge in bis(silyl)amide ligand. In a previous work (CCS Chem. 2021, 3, 388), it was predicted that the bent cationic two-coordinate model [Dy(DBP)2]⁺ (HDBP = 2,6-di-tert-butylphenol) with an O-Dy-O angle of only 155° possesses an effective energy barrier approaching ca. 2300 cm⁻¹, indicating the great potential of bulky monodentate phenolate in inducing enormous CF splitting. This closely related work should be discussed.

2. In addition to the nature of ionic bonding between Ln(III) ions and ligands, the large ionic radius and the variable coordination patterns of Ln(III) ions also bring many difficulties in constructing linear coordination geometry. Therefore, the sentence "this is difficult to achieve for lanthanide complexes as they tend to maximise coordination numbers due to their ionic bonding" should be revised.

3. The authors point out that the diluted complex, 5%Dy@1-Y, was prepared by co-crystallisation of a mixture of pure Dy species and pure Y species. However, the precise doping ratio of the test sample should be determined by inductively coupled plasma (ICP) measurements.
4. In the reaction between LnI3 and $[K\{N(SiPr_3)_2\}]$, an in situ dehydrogenative C-C bond rearrangement of the ligand scaffold has occurred. I believe that the authors have figured out the mechanism of this reaction, and I also understand the statement "The mechanism of this transformation will be elucidated in a forthcoming separate study", but I think a concise description about the mechanism is necessary in this paper.
5. As Figure 1b represents the entire molecular structure of 1-Dy, the sentence "Molecular structure of the cation of 1-Dy at 99.96(18) K..." in the caption of Figure 1 should be revised. In addition, in Figure 2b, it is the purple rather than the pink solid line represents the sum, right? Please check it.
6. In the conclusion, the authors state that the stronger magnetic anisotropy combined with the rigid ligand is expected to reduce low-energy spectral density and hence suppress two-phonon Raman relaxation. This referee agrees with this perspective but it is necessary first to elucidate the relationship between the effective energy barrier and the blocking temperature (see J. Am. Chem. Soc. 2024, 146, 18899). In addition, a strategy to further improve the blocking temperature of bis-cyclopentadienyl dysprosocenium SMMs was also proposed, i.e. combining bulky monodentate phenolate with Cp ligand to construct strong magnetic anisotropy and rigid half-sandwich dysprosium metallocenes (see Natl. Sci. Rev. 2022, 9, nwac194). Such complexes are expected to have large CF splitting and the suppressed Raman relaxation process benefitting the strong interaction between Ln(III) ions and phenolates and unique vibrational modes of the Cp ligand, respectively. Therefore, these two papers should be discussed and the conclusion may be further revised.
7. As the authors mention, the magnetic relaxation rates of 1-Dy don't switch from the slower Raman regime to the faster Orbach regime until 120 K, it is better to provide the value of t_{switch} to confirm this (see Chem. Commun. 2019, 55, 7025).

Version 1:

Reviewer comments:

Referee #3

(Remarks to the Author)

I appreciate the efforts made by the authors to address all my comments and concerns. I am satisfied with the changes made to the manuscript. It is well-known that the synthesis of a dysprosium bis(amide)-alkene complex has always been challenging in the fields of coordination chemistry and molecular magnetism. In this work, the authors achieve such goal and provide thoughtful and comprehensive magnetic analysis. In particular, they have successfully performed ab initio calculations of the magnetic hysteresis loops in such complex, which will play a significant role in elucidating the essential relationship between effective energy barrier and the blocking temperature. The manuscript has undergone considerable improvements, and, in my opinion, it merits publication in Nature.

Referee #4

(Remarks to the Author)

Emerson-King et al. presented a new dysprosium amide-alkene complex with remarkable magnetic properties. I have thoroughly reviewed the paper, supplementary information (SI), and the authors' rebuttal letter. In my opinion, the paper demonstrates sufficient novelty to warrant publication, though it requires major revisions.

The most relevant critiques from the previous referees can be grouped into three categories:

1. Sample Purity: I believe the authors have provided adequate evidence to confirm the sample's purity. Importantly, the observed properties cannot be attributed to impurities. This can be verified by comparing the ac isothermal susceptibility with the chi dc. For instance, at 100 K, the chi isothermal is approximately 0.12, while chi dc is around 0.135, indicating that a significant portion (about 90%) of the magnetic ions contributes to the slow relaxation. A brief comment on this point could clear up any misunderstandings.
2. Reaction Mechanism Not Explained: While the decision not to discuss the reaction mechanism is debatable, I agree with the first referee's view that the clear and urgent chemical message of the paper justifies this choice. Overall, I align with the first referee's perspective on this matter.
3. Hysteresis: The authors attempted to simulate the magnetic hysteresis. Although the simulation is not perfect, the effort is commendable and easily reproducible, making it appealing to a broad scientific audience. That said, I have a few suggestions that could improve the clarity of the paper.

Main Comments:

- Several figures in the SI are not referenced in the main paper, and some are not even cited within the SI itself. This results in a paper that is accessible to non-experts but leaves important information for field specialists buried in the SI or inadequately explained. I recommend moving some of this information (see after) into the main text. I am confident the authors can find a concise way to incorporate these details, keeping the typical short format of Nature.
- The dc magnetic properties are not discussed in the main text. While they may not be the primary focus, the fact that they have not been fully reproduced is an important consideration. The chi vs T graph in the SI (Figure S36) is not mentioned in the main text, along with several other SI figures. Specifically, the ab initio curve is nearly linear, which aligns with the predicted extreme axiality. Although a deviation of less than 10% at high temperatures is acceptable, the discrepancy at low temperatures suggests something is missing from the calculations. The dilution effect in the diamagnetic analogue alters the low-temperature chi, hinting at a relevant role of the interactions. I suggest explicitly stating that the discrepancy between ab initio and experimental results at low temperatures is notable. This information is essential for providing an unbiased view of the work. Many of the authors' conclusions (e.g., tunneling probabilities) depend heavily on the accuracy of the ab initio calculations.

- How did the authors determine the gamma linewidth for the T1 simulations? Including a graph showing different values could help assess the reliability of the chosen value.
- To accurately simulate the direct mechanism, high precision on low-temperature phonons is critical. Many functionals can yield varying results. Have the authors tested multiple functionals? What is the impact of these variations on the results?

Minor Comments:

- Please remove the word "recently" from the beginning of the abstract.
- Reference 7 should be removed, as the sentence refers only to reference 6.
- On page 2, line 34, the phrase "isolated by repeat [...]" needs clarification: how many repetitions were performed?
- The color scale in Figures 2a or 2d should be adjusted to avoid misinterpretation. The green and violet colors have different meanings in the two panels. I suggest changing panel 2a, as panels 1d and 3a are consistent.
- Figure S41 has the potential to be very informative, but in its current form, it is too crowded. Removing some curves could improve clarity and provide a better estimation of the relevant parameters.

Version 2:

Reviewer comments:

Referee #4

(Remarks to the Author)

I am satisfied with the work done by the authors. I believe that the manuscript is now clearer, and I am happy to recommend publication, provided that the authors correctly describe the distribution bars reported in the graphs (main text) in the corresponding captions, in accordance with Nature's policy.

We thank all referees for reviewing our paper, and to the Nature editorial team for handling this submission. We address all comments below, and we have made changes to the paper to reflect these responses. The remarks of Referee 2 have been rebutted in full, and the comments of Referees 1 and 3 have also been fully addressed. Most importantly for Referee 1, we have now performed the first ever *ab initio* calculations of the magnetic hysteresis loops in SMMs, which show excellent agreement with experiment and explain the origin of the behaviour in this new material beyond doubt. Together with improved context and additional signposting, we present that the novelty of this work is now far more evident to a generalist reader.

Referee #1 (Remarks to the Author):

The manuscript reports on the new record magnetic anisotropy of easy axis type in a Dysprosium complex with almost linear coordination and, consequently, the record barrier for the reversal of the magnetization as a Single-Molecule Magnet.

The work highlights how the current understanding of the correlation between molecular structure and magnetic properties can guide the synthetic design to increase the performances of these materials.

The work is solid, comprehensive, and clearly presented. The authors have made the choice to leave the investigation of the mechanism of reaction to a separate publication. This choice is probably fine if Nature is the journal where the manuscript will be published but should be reconsidered if the manuscript is published in a more specialist journal.

We thank the Referee for their thorough and constructive review of this paper. We are still working on the separate manuscript covering the mechanism of formation of the allyl complex. This analysis is not straightforward and has required a separate dedicated study in order to isolate intermediates by different synthetic routes. We have also synthesised multiple novel complexes not in the present paper to extract kinetic and thermodynamic parameters via NMR spectroscopy, and we have had to perform reaction pathway modelling by DFT calculations. Together this work is forming a separate full paper to be submitted as a standalone work.

Concerning the significance, I have found that the title is misleading, and, in my opinion, it should be changed. It is not clear what the authors want to communicate with 'soft hysteresis'. The two terms are somehow in conflict, as for a magnet the hysteresis is associated with the 'hardness' of the material. Thus, I was expecting some counterintuitive and completely new mechanism or phenomenon connected to hysteresis. Most likely the authors refer to the almost negligible coercive field at high temperature. However, no efforts were devoted to simulating the hysteresis. Thus, it is not granted that the property emphasized in the manuscript ($H_c=5$ Oe at 102 K, Figure 2D) is fully rationalized, in particular the flattening that is observed above 100 K in Figure 2D.

We thank the Referee for this excellent suggestion. We note that *ab initio* calculations of magnetic hysteresis loops, accounting for the explicit phonon modes and the spin-orbit coupling of the specific molecular crystal, have never been reported previously. Gratifyingly, we have successfully performed such an *ab initio* simulation of the magnetic hysteresis for complex **1-Dy** in this revision, on the basis of spin dynamics calculations as a function of magnetic field strength, field orientation, and temperature, achieving excellent agreement with experimental data. We have expanded this study to also include simulations of the magnetic hysteresis for a high-temperature Dy Cp^R SMM, as a direct comparison. This allows us to confirm, through theory, that whilst hysteresis loops of other high-performing SMMs have larger coercive fields than **1-Dy** at low temperature, **1-Dy** maintains hysteresis with a small coercive field to much higher temperatures, and shows a much slower closing of the hysteresis loops. This property arises due to the onset of the Orbach relaxation

mechanism at higher temperature, and is fully-supported by our field-dependent relaxation calculations, thus rationalising the very small coercive fields at higher temperature and justifying the title of the paper. We have added an explanation of the term “soft hysteresis” to mean hysteresis with small coercive fields and small remanent magnetisation to the discussion, and have added figures showing the remanent magnetisation to the supporting information. We also clarify that the flattening above 100 K corresponds to the hysteresis loops being closed within error. Perhaps even more importantly, the achievement of a first-principles calculation of magnetic hysteresis represents an independent significant research milestone for the research field, beyond the synthetic achievement and experimental observation we presented in the originally submitted manuscript. We sincerely thank the Referee for this suggestion – new achievement unlocked!

This new SMM exhibits the record barrier for magnetization reversal, but the under-barrier mechanisms (Raman or QTM) are more efficient than in other reported high-temperature SMMs. As a result, the relaxation time levels at low temperatures to a value that is even lower than the 100 s considered in magnetism textbooks as the threshold for magnetization blocking (see Figure 3).

The authors are crystal clear about the fact that they have succeeded in improving one parameter, magnetic anisotropy, by switching from sandwich complexes to linear ones, at the expense of the others, i.e., the efficiency of low-energy phonons and QTM.

We did not previously discuss the lack of a 100 s blocking temperature in **1-Dy**, but upon reflection from this comment we agree it is useful to discuss this point explicitly. We now make it clear in the paper that the T_{B100s} is far inferior to the values that have been achieved by axial Dy Cp^R complexes. However, we contend that the drastically-different high-temperature behaviour is a very important observation for this research field. Indeed, as can be seen in references 8-18 (and the related literature on other high-temperature SMMs), the vast majority of the focus on SMM properties is on the temperatures at which open magnetic hysteresis is observed and not T_{B100s} values. An important point is that we show the differences between the magnetic behaviour of **1-Dy** and other high-performance Dy Cp^R SMMs, where it outperforms all other SMMs by both U_{eff} and magnetic hysteresis metrics. There is plenty of scope to improve both these values, and T_{B100s} , further, using the methodology we present. Hence, the present work suggests there are many more exciting results ahead, far beyond the plateau that appears to have been reached for Dy Cp^R SMMs.

As a result, the manuscript is borderline for acceptance in one of the most generalistic journals. On the contrary, the take-home message on chemical design is very strong. We thank the Referee for their review, which we note is overall very positive but has erred on the side of not recommending acceptance in this journal. Upon reflection we feel that we did not convey the novelty aspects of this paper well enough to a wide audience, and in particular readers more interested in the physical properties of SMMs rather than the synthetic advances presented herein. The comments of this Referee have been invaluable for pointing out where we had missed opportunities to provide context, and we humbly suggest that the addition of the first-principles modelling of the magnetic hysteresis, which fully explains the experimental results, provides additional excitement for the general reader.

I have also some technical comments:

- Please, clearly state in the main text how "stable" is the material and how it must be handled.

We now state in the main text that “All compounds reported herein are highly air- and moisture-sensitive and must be handled under inert atmospheres.”

- Synthesis of 5%Dy@1-Y: Is really 2-Dy mixed with 1-Y, or is it just a typo?
This typo has been corrected.

- X-ray powder diffraction: The authors should say if the broad peaks come from the sample holder or from an amorphous phase of the compounds

We now state the reason in the revised SI: "The two broad peaks centred around approximately 16° and 42° 2θ are due to scatter from the Fomblin YR-1800 oil used to mount the sample and protect it from atmospheric degradation." We note that we are somewhat limited as to how samples are prepared for PXRD analysis by the high air- and moisture-sensitivity of the complexes studied.

- Figure S57. The data fitting at the lowest temperature is very poor, but the alpha parameter and its error are reasonable.

We have looked again at the lowest temperature dataset, and noted that the waveform moment vs time data is very noisy, and so we agree that an accurate determination of the relaxation time cannot be made. This is due to frictional heating which occurs at the lowest temperatures when measuring in VSM mode for some samples. We have therefore removed the 2.5 K data on 5%Dy@1-Y from the paper, but our overall interpretation remains unchanged, as the 4 K and 7.5 K data show a plateauing of the relaxation rate akin to the pure sample.

- When discussing the results of ab initio calculations the authors refer to Table S6. However, in the SI there are 4 of these tables with slightly different values among them and with the values reported in the main. A comment about the different models used to obtain Tables S6-S9 and the criteria employed to assess which one is the most accurate should be given in the main text.

We have now indicated in the text that we consider Table S6 to be the most representative as it reports calculations on the major crystallographic component; however, the electronic structure should be considered as a range between the major and minor components (Table S7). We have also corrected the values in the text to specifically refer to the major component.

In our modelling of the relaxation rates and hysteresis, we require a solid state DFT-optimised geometry to obtain phonons. This optimised geometry has a more bent N-Dy-N angle than the crystal structure, resulting in a reduced barrier. To model the spin-phonon coupling we include point charges to represent the infinite periodic potential, and to reduce computational cost of the spin-phonon calculation, we reduce the number of states in the CASSCF wavefunction (Table S10). Because of this discrepancy in methodology, to assess the effect of the change in geometry we also performed a CASSCF calculation at the DFT geometry with methodology matching that of calculations on the crystal structure geometries. We have reorganised the computational methodology to make this distinction clearer.

Referee #2 (Remarks to the Author):

This work cannot be published in any journal due to the poor experimental quality of the reported compounds.

We outright refute this assessment. The points made by this Referee regarding the quality of characterisation data have been presented as if they are factual, but they are all, in fact, subjective opinions which do not stand up to scientific scrutiny or literature standards. We rebut each of these comments fully, explaining why they are all factually incorrect and entirely subjective. We also note that this Referee did not review or comment upon the scientific content of the paper beyond the characterisation data, which does not provide us with much constructive criticism to improve this work.

1. The proof of purity for all new compounds is highly questionable.
The carbon values of the elemental analyses for all compounds are:

1-Y: expected 36.49, found 33.27.
1-Dy: expected 34.99, found 32.92.
5%Dy@1-Y: expected 36.42, found 33.47,
2-Y 58.17, found 55.74.
2-Dy 52.93, found 52.80.

It is obvious that the elemental analyses are far off from the expected values. In fact, discrepancies in C values (calc vs found) are 3.22% (1-Y), 3.52% (1-Dy), 2.95% (5%Dy@1-Y), 2.43% 2-Y, 0.13% 2-Dy.

We strongly disagree with the statement that the proof of purity of new compounds in this paper is “highly questionable.” We agree that elemental analysis (EA) results obtained herein often deviate from expected values, and indeed we had already commented on this in the SI.

It is not prudent to view EA data in isolation for the following reasons:

1. EA results are very often not in good agreement with predicted values. A cursory glance of the literature will reveal countless EA results that do not fit within the arbitrary limits set by journals (including *Nature*) as “acceptable.” This is the first issue with the comment made above: the Referee is attempting to treat us differently to other authors, and to hold us to a standard that is above that of, and inconsistent with, the existing scientific literature.
2. It has been discussed in multiple journal articles and editorials that using EA as an absolute standard of purity is a falsehood. A recent international study (*ACS Cent. Sci.*, 2022, **8**, 855) shows that EA results vary wildly even on pure standard samples between different runs at the same facility, and that it is statistically likely to get results outside the arbitrary “acceptable” range; we now cite this paper in the revision. We hope the Referee would agree with us that collecting EA data until you get a “good” result, and then not reporting all the other obtained results that didn’t fall within range, is scientifically indefensible. EA results are well-known to be worse for highly air-sensitive organometallic complexes, where carbide formation often leads to lower %C values than expected (see refs. 2 and 5-8 in SI as examples for Ln silylamide complexes), and fluorine-rich compounds are the worst offenders for giving EA results that are not in accord with predicted values (*J. Anal. Chem.*, 2008, **63**, 1094). The present compounds are exactly in both these classes of materials, so we now cite this paper in the context of our EA results. If the bar set by the reviewer was stringently applied to all authors, then the number of papers published by organometallic chemists would conservatively shrink instantly by ca. 50%.
3. We demonstrate the bulk purity of the novel complexes reported herein holistically by a combination of other methods (overlapping ATR-IR spectra for Y/Dy pairs, multinuclear NMR data for Y analogues, pristine magnetic data for Dy analogues, good agreement of expected and found powder XRD patterns for **1-Ln**, and excellent agreement of calculated magnetic properties with experimental data). Indeed, the first-principles simulation of the magnetic hysteresis shows almost exactly the correct temperature dependence of the data sets of two compounds, which is taken entirely from the crystal structure alone. This high correlation from different methods is not a coincidence. This Referee has ignored these complementary data and has focussed solely on the EA data.
4. We note that both Referees 1 and 3 did not raise any issues with the characterisation data reported for complexes herein beyond one request for ICP data on the doped sample (which we have done, see below). The consensus is that the results presented are reliable, and that Referee 2’s opinion is an outlier.

2. It is evident from the moderate R values of all reported compounds, that the quality of the crystals is poor and, in addition, the molecular structures are highly disordered.

1-Y: R1 = 6.64, wR2 = 19.74.
1-Dy: R1 = 9.03, wR2 = 29.19.
5%Dy@1-Y: R1 = 9.38, wR2 = 30.36.
2-Y: R1 = 7.18, wR2 = 17.92.
2-Dy: R1 = 10.70, wR2 = 25.56.

Having said this and keeping in mind that magnetism is extremely sensitive to composition (impurity), any claim for a magneto-structural relationship is obsolete.

We reject the assertion that the crystallographic data are not of high enough quality to make reliable comments on the magnetic data of the complexes herein. We contend that the data reported are reliable and are within the literature standard. We have already been clear in the paper about the disorder of the cations in **1-Ln** and the corresponding extent to which magneto-structural correlations can be made. Again, the Referee has ignored complementary data, such as the NMR data of the yttrium analogues clearly showing the structures obtained from the solid-state XRD experiments are indeed correct. They have also ignored that our calculations – based solely on the crystal structure – are in excellent agreement with the bulk magnetic measurements. As **2-Ln** are precursors to **1-Ln**, and their identity has been reliably established, the connectivity in each of these datasets is clear-cut. The Referee is basing their assessment of single crystal XRD data solely on R values, which is too simplistic an approach to use to appraise crystallographic data - an appreciation of the nuances of complex experimental data is crucial for correct interpretation, in line with the comments presented above on elemental analysis.

Three of the compounds in question (**1-Y**, **1-Dy** and **5%Dy@1-Y**) contain the weakly coordinating anion (WCA) $[Al\{OC(CF_3)_3\}_4]^-$, which has well-known issues with crystallographic disorder (for example, see refs#13, 18 and 25 of the manuscript). This exacerbates any issues associated with disorder of the cations, and both **2-Y** and **2-Dy** share these issues due to the flexibility of ligands in the metal coordination spheres. The difficulty in handling these air-sensitive samples ostensibly rules out the use of a synchrotron source to collect higher resolution data. However, such an endeavour would not elucidate any additional useful chemical information about the structure, other than the degree of disorder of the anion, which is not the focus of what we are extracting from the data: it is the connectivity and metrical parameters of the cations that are of most interest. We would gladly have a dedicated crystallographic referee review our data. To address this comment in full we have added the following discussion and figures of electron density plots to the SI:

“The $\{OC(CF_3)_3\}$ substituents of the $[Al\{OC(CF_3)_3\}_4]^-$ WCAs are highly disordered in the datasets for **1-Y**, **1-Dy** and **5%Dy@1-Y**, which strongly influences the statistics for the fit of the models (i.e. leads to larger R_1 and wR_2), especially to the weaker higher angle data. This issue is in addition to the disorder of the silylamide-containing components, which occur due to the intrinsic flexibility of the metal coordination spheres, giving a wide range of metrical parameters. The disorder in these systems cannot be fully modelled while retaining a suitable data to parameter ratio for the given resolution of the datasets, making the overall fit to the data relatively poor compared to systems without such disorder. This is a known problem for crystal systems containing this particular WCA and related examples such as $[Al\{OC(CF_3)_3\}_4]^-$, as evidenced by papers cited in the manuscript (refs#13, 18 and 25). These previously reported datasets generally have high R_1 and wR_2 values, even when the data are modelled as effectively as possible and low goodness of fit metrics are obtained. Plots of electron density maps (Figures S22–S31) derived from the data show that the phases derived from the models fit well for the observed structure factors. Plots of $2F_o - F_c$ show good agreement of the model of the main molecule with the electron density map, with the majority of deficiencies around the WCAs, confirmed by plots of $F_o - F_c$ difference maps.

This indicates that the model is appropriate, especially for the cations of interest, for the given resolution and therefore size of the dataset. The noise around the heavy metal sites in the F_o-F_c difference maps is common when collecting data for compounds containing heavy metals using $\text{CuK}\alpha$ radiation; the small size of the crystals precluded the use of a lower brilliance, shorter wavelength, $\text{MoK}\alpha$ source when collecting the data. This also limits the maximum resolution of the dataset that can be obtained, limiting the extent that disorder of the WCAs can be modelled to improve the fit.”

3. Another concern refers to the discussion of the metal coordination features of the compounds which is quite confusing: the coordination mode of the metal centers is clearly NOT linear. Apparently, the compounds are diamido pi olefin-metal complexes by no means linear but containing four-coordinate metal centers. Any reasonable textbook (e.g., Cotton-Wilkinson) will tell and falsify this inappropriate and artificial description.

We are somewhat confused by this comment as we do not believe that we suggested that any of the complexes in this paper are linear. During the discussion of the single crystal XRD data we referred to the N-Dy-N angle of **1-Dy** being made “more linear” by a combination of alkene coordination, dispersion and crystal packing forces. This sentence clearly refers to a single interatomic bond angle rather than the overall complex geometry. However, as we would not want any reader thinking that we are making “inappropriate and artificial” descriptions of the geometry, we have changed the descriptor to “less bent.” Similarly, in the subsequent paragraph on CASSCF calculations on the XRD-determined structure of **1-Dy** we referred to a “bent molecular geometry”, whereas this has now been changed to “the bent N–Dy–N angle” to remove any potential confusion. Finally, in the final paragraph of the paper we suggested that if in future other authors were able to make Dy complexes with “more linear coordination geometries” then they would achieve better magnetic properties than the complex presented herein. Again, our intention was not to suggest that we had achieved a linear coordination geometry in our complex, so to remove any potential for misinterpretation, we now suggest that others target similar Dy complexes with “more linear E–Dy–E angles (i.e. $> 150^\circ$; E = monodentate donor atom).

The coordination number (CN) of alkene complexes is actually not as straightforward as referring to Cotton & Wilkinson as the Referee has indicated. Indeed, the description in this textbook is contrary to IUPAC guidelines, which state that CNs should only be strictly used for σ -donor and not for multihapto-ligands (<https://goldbook.iupac.org/terms/view/C01331>). An extended discussion of these arguments is not warranted here and would be beyond the scope of this paper. We note that although we had therefore been careful to avoid referring to the CN of **1-Dy** throughout the paper we did refer to it as formally two-coordinate in the conclusion of the originally submitted manuscript. We have now removed this claim and any controversy that it may have inadvertently caused.

4. There are many self-citations, and other significant references are not mentioned.

We find this criticism somewhat unusual, and the lack of detail provided here does not give us sufficient guidance on how the Referee would intend for us to address their concerns. The submitted paper contained 47 references, which is close to the upper limit of 50 following the *Nature* guidelines. It is therefore natural that not all relevant literature will be cited, but we made our best efforts to cite what we believed to be the most significant examples. If this Referee had provided suggestions for how best to fill up the remaining three slots in the reference list then we could have considered them, but we note that Referee 3 has requested that we add three citations. Amongst these 47 original references, there are a relatively low number of self-citations for the two corresponding authors, especially when it is considered that as a collaborative team they have made a large number

of high-performance Dy SMMs, as well as Ln bis-amide complexes that have provided the inspiration and concepts that underpin this work.

There are currently ten citations to Mills, of which one is where they are co-editor of a recent general f-element textbook, two are relevant book chapters (one on Ln silylamide chemistry and one on low-coordination number f-element complexes), two are on two-coordinate Ln bis(silylamide) complexes, and five are on the synthesis and relaxation mechanisms of the highest-performing Dy SMMs to date. The additional citations to Chilton are due to collaborations with multiple other synthetic SMM research groups and the development of methods that are used to analyse high-performance Ln SMMs. There are many relevant works by the corresponding authors that we have not cited in an effort to balance the citation list with the most relevant literature from other authors; several of these have been cited in the SI only. We contend that the presence of these citations in the reference list are all fully justifiable and we refute the suggestion that citing our most relevant work is not an acceptable thing to do.

Again, the insufficient quality of the central experimental data and the confusing structural discussion disqualifies the paper to be published in any scientific journal.

We have rebutted all points made by this Referee in full regarding the quality of the experimental data presented and the structural discussion. We respectfully contend that the paper is suitable for publication and we reject the suggestion that our work should be disqualified from publication arbitrarily and permanently by this Referee.

Referee #3 (Remarks to the Author):

In the field of single-molecule magnets (SMMs), constructing linear two-coordinate trivalent lanthanide (Ln) complexes has long been a dream but is a formidable challenge from the perspective of chemical synthesis due to the large ionic radius and variable coordination pattern of Ln(III) ions. In general, the monodentate ligand with bulky or even super-bulky side groups can serve as an ideal platform to achieve such goal owing to the electrostatic interaction between Ln(III) and ligand. In this work, Mills et al. reported a dysprosium bis(amide)-alkene complex, 1-Dy, featuring relatively short Dy-N bond lengths and a slightly bent N-Dy-N angle at the axial position, and weak Dy-C interactions in the equatorial plane. Although X-ray crystallography demonstrated that this compound doesn't have the strictly two-coordinated linear arrangement, the weak bonding between lanthanide and proximal atoms in the equatorial plane arising from Dy-Calkene and Agostic-type interactions plays a vital role in stabilizing solid-state structures. More importantly, the transverse crystal field (CF) induced by above weak interactions does not significantly diminish axial CF resulting from the strong Dy-N bonding, which was clearly proven by *ab initio* calculations and QTAIM analysis of isostructural Y(III) compound. The complex exhibits the largest effective energy barrier and the highest blocking temperature for all known SMMs, i.e. 1843 cm⁻¹ and 100 K. The detailed theoretical calculations, including static and dynamic CF considerations, have also been carried out, elucidating the relaxation mechanism of magnetization in 1-Dy. Without doubt, this work reported a remarkable advance in SMM field, and this is going to be a milestone in this direction. I believe it will appeal to a wide cross-section of the chemistry community, including synthetic coordination and organometallic chemists, theoretical chemists as well as materials chemists, and will be cited by every researcher in SMM field. Therefore, I am delighted to recommend acceptance of the manuscript after addressing the following points:

We thank the Referee for their overwhelmingly positive review of this work, and for the constructive review provided.

1. Theoretical calculations show that the SMM performance of $[\text{Dy}(\text{C}_5\text{iPr}_5)(\text{C}_5\text{Me}_5)][\text{B}(\text{C}_6\text{F}_5)_4]$ has reached the upper limit of $[\text{Dy}(\text{CpR})_2]^+$ family. In this work, the authors ascribed the larger effective energy barrier of the dysprosium bis(amide)-alkene complex than that of $[\text{Dy}(\text{C}_5\text{iPr}_5)(\text{C}_5\text{Me}_5)][\text{B}(\text{C}_6\text{F}_5)_4]$ to the shorter Dy-N distance and the more dense negative charge in bis(silyl)amide ligand. In a previous work (CCS Chem. 2021, 3, 388), it was predicted that the bent cationic two-coordinate model $[\text{Dy}(\text{DBP})_2]^+$ (HDBP = 2,6-di-tert-butylphenol) with an O–Dy–O angle of only 155° possesses an effective energy barrier approaching ca. 2300 cm^{-1} , indicating the great potential of bulky monodentate phenolate in inducing enormous CF splitting. This closely related work should be discussed.

We thank the Referee for bringing this paper to our attention. We now discuss and cite this work in the revised paper as suggested. However, it is prudent to discuss this work in the context of previous work performed on theoretical near-linear Dy(III) bis-(amide), -(alkyl) and -(methanediide) complexes, which were predicted to show U_{eff} up to 3000 cm^{-1} , so we have also added these citations and discussion to the revised manuscript.

2. In addition to the nature of ionic bonding between Ln(III) ions and ligands, the large ionic radius and the variable coordination patterns of Ln(III) ions also bring many difficulties in constructing linear coordination geometry. Therefore, the sentence “this is difficult to achieve for lanthanide complexes as they tend to maximise coordination numbers due to their ionic bonding” should be revised.

We agree that we should have elaborated further here to make these aspects clear to the generalist reader. The revised paper now incorporates these additional recommended points.

3. The authors point out that the diluted complex, 5%Dy@1-Y, was prepared by co-crystallisation of a mixture of pure Dy species and pure Y species. However, the precise doping ratio of the test sample should be determined by inductively coupled plasma (ICP) measurements.

We performed ICP-MS on this sample as requested, finding 7% Dy : 93% Y. However, the magnetic data obtained are more in accord with a ratio of 3% Dy : 97 % Y. Given the experimental uncertainties involved in both of these measurements, we now state in the revised paper that the doped sample is 5(2)% Dy and 95(2)% Y.

4. In the reaction between LnI_3 and $[\text{K}\{\text{N}(\text{SiiPr}_3)_2\}]$, an in situ dehydrogenative C-C bond rearrangement of the ligand scaffold has occurred. I believe that the authors have figured out the mechanism of this reaction, and I also understand the statement “The mechanism of this transformation will be elucidated in a forthcoming separate study”, but I think a concise description about the mechanism is necessary in this paper.

Working out the mechanism of formation of the allyl complex has proved to be non-trivial (see response to Referee 1 above). The main issue preventing us from commenting on this in more detail in this paper is that there are no data included herein that would allow us to reasonably speculate further. Indeed, crucial kinetic and thermodynamic parameters, as well as reaction intermediates, can only be extracted if **1-Ln** is made by an entirely different synthetic route to the one we report here. Further, the mechanism that we are currently elucidating would not necessarily be identical to the reported reaction. The concise description of “dehydrogenative C-C bond rearrangement” is therefore at the limit of what are able to confidently propose here.

5. As Figure 1b represents the entire molecular structure of 1-Dy, the sentence “Molecular structure of the cation of 1-Dy at 99.96(18) K...” in the caption of Figure 1 should be revised.

In addition, in Figure 2b, it is the purple rather than the pink solid line represents the sum, right? Please check it.

We have corrected the caption of Figure 1, and Figure 2b correctly shows the sum fit as a solid pink line.

6. In the conclusion, the authors state that the stronger magnetic anisotropy combined with the rigid ligand is expected to reduce low-energy spectral density and hence suppress two-phonon Raman relaxation. This referee agrees with this perspective but it is necessary first to elucidate the relationship between the effective energy barrier and the blocking temperature (see *J. Am. Chem. Soc.* 2024, 146, 18899). In addition, a strategy to further improve the blocking temperature of bis-cyclopentadienyl dysprosocenium SMMs was also proposed, i.e. combining bulky monodentate phenolate with Cp ligand to construct strong magnetic anisotropy and rigid half-sandwich dysprosium metallocenes (see *Natl. Sci. Rev.* 2022, 9, nwac194). Such complexes are expected to have large CF splitting and the suppressed Raman relaxation process benefitting the strong interaction between Ln(III) ions and phenolates and unique vibrational modes of the Cp ligand, respectively. Therefore, these two papers should be discussed and the conclusion may be further revised.

We agree that the relationship between effective energy barrier and the blocking temperature needs to be better-defined. However, we do not understand the comment about the effective energy barrier's relationship to the blocking temperature: the cited paper by the Referee is about a very nice di-dysprosium complex that shows open hysteresis and suppression of quantum tunnelling of the magnetisation due to antiferromagnetic exchange bias, so while this is lovely work, we do not see the relevance for monometallic SMMs such as the present compound. Hence, we politely decline the request to cite this work here. By addressing the comments of Referee 1 to model the hysteresis loops we now believe that we have achieved this aim for the complexes herein, and we have also shown the community how this can be achieved for the complexes that they make themselves.

We thank the Referee for bringing this Perspective article to our attention regarding the potential of mixed Ln SMMs with one aromatic ligand and one monodentate ligand. This is a very relevant work and we have added this citation to the paper and discuss it in the conclusion.

7. As the authors mention, the magnetic relaxation rates of 1-Dy don't switch from the slower Raman regime to the faster Orbach regime until 120 K, it is better to provide the value of τ_{switch} to confirm this (see *Chem. Commun.* 2019, 55, 7025).

We thank the Referee for the good suggestion: we now give a value of $\tau_{\text{switch}} = 6.7$ s, which occurs at 105 K.

We thank the referees for reviewing our paper, and to the Nature editorial team for handling this submission. We are delighted that the reviewers are highly supportive of publication of this work in this journal. We address all comments from the second round of review below, and we have made changes to the paper to reflect these responses.

Editorial comments:

1. Overall structure: Although it is not essential that every item of supplementary information is cited in the main paper, the rationale for its inclusion in the supplementary information should nevertheless be clear – don't simply load it with uncommented information that is not referred to anywhere (even in the supplementary information!). You might also like to consider making use of our Methods and Extended Data formats for increasing the visibility of the most important items of supplementary information – these will then be integrated into the final version of the paper. (We allow ~3,000 words of Methods, and up to 10 items of Extended Data. More details below.)

As advised by the editor, we have added a Methods section of ~3000 words and ten items of Extended Data to the paper. Details that are of secondary importance remain in the supplementary information, but now all figures and tables are referred to in the main text so the rationale for their inclusion is now clear. This required a significant reorganisation and rephrasing of the material in the paper and supplementary information.

2. Length of the main paper. With about 2,700 words (body text, excluding abstract) and 3 figures (two of them large), the paper is already pushing at our regular length limits. So please try to ensure that the main paper does not get much longer in this iteration (a Methods section could help here too).

We have addressed the referee and editorial comments, and we have added a reference to another Dy bis-amide SMM paper that was published recently. These changes increased the length of the revised paper, beyond Nature guidelines (from ~2700 to ~3000 words for the main body of the paper, excluding abstract, figure captions, references and various author statements; guidelines state ~2500 words), and the number of main paper references to 52. To address these issues we have contracted the size of the introduction and conclusion following journal guidelines and we performed some edits to the paper throughout to increase word economy. This now makes the article ~2500 words and the Methods ~3000 words, which are both within allowable limits.

3. Please note the specific instructions for chemical structure presentation.

Fig.1 has been modified to align with the Nature guidelines.

Referee #3:

I appreciate the efforts made by the authors to address all my comments and concerns. I am satisfied with the changes made to the manuscript. It is well-known that the synthesis of a dysprosium bis(amide)-alkene complex has always been challenging in the fields of coordination chemistry and molecular magnetism. In this work, the authors achieve such goal and provide thoughtful and comprehensive magnetic analysis. In particular, they have successfully performed ab initio calculations of the magnetic hysteresis loops in such complex, which will play a significant role in elucidating the essential relationship between effective energy barrier and the blocking temperature. The manuscript has undergone considerable improvements, and, in my opinion, it merits publication in Nature.

We thank the referee for reviewing this paper a second time, and for their positive appraisal of the improvements that we have made to the article following the advice of reviewers to make it now suitable for publication. There are no points to address here.

Referee #4:

Emerson-King et al. presented a new dysprosium amide-alkene complex with remarkable magnetic properties. I have thoroughly reviewed the paper, supplementary information (SI), and the authors' rebuttal letter. In my opinion, the paper demonstrates sufficient novelty to warrant publication, though it requires major revisions.

We thank the referee for their thorough and comprehensive review. We are pleased that they are supportive in principle of publication in this journal. We note that the description of "major revisions" is regarding a recommendation to reorganise how results are presented in the paper and SI rather than major scientific changes, and the changes that were required to achieve this were actually relatively minor. We address all comments in full below.

The most relevant critiques from the previous referees can be grouped into three categories:

1. Sample Purity: I believe the authors have provided adequate evidence to confirm the sample's purity. Importantly, the observed properties cannot be attributed to impurities. This can be verified by comparing the ac isothermal susceptibility with the chi dc. For instance, at 100 K, the chi isothermal is approximately 0.12, while chi dc is around 0.135, indicating that a significant portion (about 90%) of the magnetic ions contributes to the slow relaxation. A brief comment on this point could clear up any misunderstandings.

The referee agrees that the purities of samples in this paper have been proven, and that the magnetic data presented cannot be attributed to impurities.

We thank the referee for highlighting this point: while investigating, we found a small error in the plotting of the χ' data and in our treatment of the DC susceptibility data shape correction, which have now been corrected. We have also introduced corrections for the NMR tube, the eicosane background, and the diamagnetic contribution of the sample (which had only previously been done for the DC data). The revised ac χ' value at 100 K and 0.1 Hz is $0.125 \text{ cm}^3 \text{ mol}^{-1}$ and the field-cooled DC susceptibility is still $0.135 \text{ cm}^3 \text{ mol}^{-1}$. These values differ because there is still a significant out-of-phase component at 0.1 Hz ($0.018 \text{ cm}^3 \text{ mol}^{-1}$) and χ' has not yet plateaued (which occurs at lower frequencies and out of range of our instrumentation). Our modelling accounts for this, and the revised value of the isothermal susceptibility is $0.136 \text{ cm}^3 \text{ mol}^{-1}$, in excellent agreement with the field-cooled DC susceptibility. We have added a comment noting the close agreement of AC and DC data, indicating that all of the Dy ions present contribute to the slow relaxation, in agreement with high sample purity.

2. Reaction Mechanism Not Explained: While the decision not to discuss the reaction mechanism is debatable, I agree with the first referee's view that the clear and urgent chemical message of the paper justifies this choice. Overall, I align with the first referee's perspective on this matter.

We note that the reviewer has come to the same consensus as both the first referee and ourselves on this matter, which is that the reaction mechanism to form **1-Dy** does not need to be determined here as it would detract from the urgent chemical message of the paper. We note that we have still not finished off our follow-up mechanistic paper, which as stated previously requires a separate investigation given its complexity. No response is required here.

3. Hysteresis: The authors attempted to simulate the magnetic hysteresis. Although the simulation is not perfect, the effort is commendable and easily reproducible, making it appealing to a broad scientific audience.

We appreciate the complimentary comments regarding the work that we have done to attempt to simulate magnetic hysteresis. No response is required here.

That said, I have a few suggestions that could improve the clarity of the paper.

Main Comments:

- Several figures in the SI are not referenced in the main paper, and some are not even cited within the SI itself. This results in a paper that is accessible to non-experts but leaves important information for field specialists buried in the SI or inadequately explained. I recommend moving some of this information (see after) into the main text. I am confident the authors can find a concise way to incorporate these details, keeping the typical short format of Nature.

We have performed a minor reorganisation of the material in the paper and SI to address the issue of how the results are presented. We have moved a limited amount of information from the SI into the main text according to the referee's instructions, noting the restrictions that we have on article length for this journal. This includes discussion of the magnetic susceptibility, zero-field cooled/field-cooled measurements, calibration with palladium and interpretation of the contributions to the loss of magnetisation from *ab initio* calculations. Most importantly, we now cite all figures and tables in both the paper and the SI as requested. As the magnetic properties of **2-Dy** are not relevant to the main message of the paper we have included all **2-Dy** characterisation only in the SI and have added a discussion therein.

- The dc magnetic properties are not discussed in the main text. While they may not be the primary focus, the fact that they have not been fully reproduced is an important consideration. The χ vs T graph in the SI (Figure S36) is not mentioned in the main text, along with several other SI figures. Specifically, the *ab initio* curve is nearly linear, which aligns with the predicted extreme axiality. Although a deviation of less than 10% at high temperatures is acceptable, the discrepancy at low temperatures suggests something is missing from the calculations. The dilution effect in the diamagnetic analogue alters the low-temperature χ , hinting at a relevant role of the interactions. I suggest explicitly stating that the discrepancy between *ab initio* and experimental results at low temperatures is notable. This information is essential for providing an unbiased view of the work. Many of the authors' conclusions (e.g., tunneling probabilities) depend heavily on the accuracy of the *ab initio* calculations.

We thank the referee for pointing out our oversight – we take for granted that readers understand all the basics, so we appreciate the reminder to keep the work accessible to a broader audience. We have made sure to comment on the DC data in the main text now. Addressing the scientific question: at low temperatures the sample shows magnetic blocking, as evidenced by the different zero-field cooled and field-cooled traces below 47 K (was Supplementary Fig. 38, now Extended Data Fig. 3a), while the *ab initio* curve is calculated for equilibrium conditions. This explains the deviation at low temperatures and is commonly observed for high-performance SMMs. We now plot the ZFC/FC susceptibility and CASSCF-SO calculated trace on the same graph to make this correlation more apparent. We have added this point to the main text, as well as commenting on the linearity of the experimental and *ab initio* curve.

- How did the authors determine the gamma linewidth for the T1 simulations? Including a graph showing different values could help assess the reliability of the chosen value.

We have added a graph of zero field rates with different linewidths to the supplementary information (Supplementary Fig. S46). We see little linewidth dependence because of the large unit cell, resulting in a dense phonon density of states. Differences are much smaller than the distribution of experimental rates (shown as error bars). We chose 10 cm^{-1} as the best compromise across Raman and Orbach mechanisms, and to be consistent with previous work. We thank the referee for suggesting this inclusion, and provide this discussion in the Methods section.

- To accurately simulate the direct mechanism, high precision on low-temperature phonons is critical. Many functionals can yield varying results. Have the authors tested multiple functionals? What is the impact of these variations on the results?

We agree this is a potential source of variation in our simulations, and have added an acknowledgement of this in the Methods section.

We chose PBE for consistency with our previous work, and as the most widely-used functional in materials chemistry. Indeed, this is actually quite an involved question that also necessitates assessment of Brillouin-zone integration and is hence tied up in the question of phonon linewidth, etc., which is a huge undertaking. We have not tested different functionals as the computational cost of performing multiple periodic geometry optimisations and phonon calculations with different functionals, and then re-performing all the spin-phonon coupling and relaxation rate calculations is not feasible for this compound with such a large unit cell. However, we are currently performing exactly such a study for a smaller molecule, where such a large and systematic theoretical study is feasible.

Addressing the scientific impact of the referee's question, it is also important to note that in our simulations, the Direct mechanism only dominates magnetic reversal at a limited range of temperatures; in **1-Dy**, up to ~15 K. In the revision, we now show the simulated coercive field including only one phonon or two phonon rates (Extended Data Figure 9c). We agree that we over-predict the Direct rates, but given the low temperature range where these dominate, compared to the higher-temperature focus of this paper, we contend that this does not adversely impact our conclusions on the observed properties.

Minor Comments:

- Please remove the word "recently" from the beginning of the abstract.

Done.

- Reference 7 should be removed, as the sentence refers only to reference 6.

Done.

- On page 2, line 34, the phrase "isolated by repeat [...]" needs clarification: how many repetitions were performed?

The removal of residual $[\text{NEt}_3\text{H}][\text{Al}\{\text{OC}(\text{CF}_3)_3\}_4]$ from the samples of **1-Dy** and **1-Y** used for the characterisation data in this paper was achieved by nine successive recrystallisations, at which point $[\text{NEt}_3\text{H}][\text{Al}\{\text{OC}(\text{CF}_3)_3\}_4]$ was no longer observed by ^1H NMR spectroscopy. A tenth recrystallisation gave pure batches of **1-Ln**. The number of repetitions required will naturally vary when the scale and operator differ, so it is not informative to give a numerical value. To address this point we have now deleted the vague word "repeat" in the main paper, and we now state more precisely in the respective experimental sections the number of recrystallisations that were performed in each case.

- The color scale in Figures 2a or 2d should be adjusted to avoid misinterpretation. The green and violet colors have different meanings in the two panels. I suggest changing panel 2a, as panels 1d and 3a are consistent.

We have updated the colour scheme in panel 2a.

- Figure S41 has the potential to be very informative, but in its current form, it is too crowded. Removing some curves could improve clarity and provide a better estimation of the relevant parameters.

We have moved Fig. S41 to Extended Data Fig. 3d. The new version is larger, shows hysteresis curves every 4 K instead of 2 K, has narrower linewidths, smaller symbol size and

does not repeat line colours. The most important parameters (coercive field and remnant magnetisation) are given in Fig. 2d and Extended Data Fig. 3c. We think it is important to report all the hysteresis data, so we have also included a full-page figure with hysteresis in 2 K steps (Supplementary Fig. 32).

We thank referee #4 for reviewing our paper one final time, and to the Nature editorial team for handling this submission. We address all comments from the final round of review and editorial guidelines below, and we have made changes to the paper to reflect these responses.

Referee #4 (Remarks to the Author):

I am satisfied with the work done by the authors. I believe that the manuscript is now clearer, and I am happy to recommend publication, provided that the authors correctly describe the distribution bars reported in the graphs (main text) in the corresponding captions, in accordance with Nature's policy.

We have now made this requested change, and we've also made analogous sentences in the captions of extended data and SI figures match this wording too for consistency.